# Interactive Adjustment for Human Trajectory Prediction with Individual Feedback

**Jianhua Sun,**\* **Yuxuan Li,**\* **Liang Chai, Cewu Lu**†
Shanghai Jiao Tong University

## Abstract

Human trajectory prediction is fundamental for autonomous driving and service robot. The research community has studied various important aspects of this task and made remarkable progress recently. However, there is an essential perspective which is not well exploited in previous research all along, namely **individual feedback**. Individual feedback exists in the sequential nature of trajectory prediction, where earlier predictions of a target can be verified over time by his ground-truth trajectories to obtain feedback which provides valuable experience for subsequent predictions on the same agent. In this paper, we show such feedback can reveal the strengths and weaknesses of the model's predictions on a specific target and heuristically guide to deliver better predictions on him. We present an **interactive adjustment network** to effectively model and leverage the feedback. This network first exploits the feedback from previous predictions to dynamically generate an adjuster which then interactively makes appropriate adjustments to current predictions for more accurate ones. We raise a novel displacement expectation loss to train this interactive architecture. Through experiments on representative prediction methods and widely-used benchmarks, we demonstrate the great value of individual feedback and the superior effectiveness of proposed interactive adjustment network.

## 1 Introduction

Human trajectory prediction is a task to forecast the future movements of pedestrians according to the observations from the past. Over the past years, researchers have studied this topic from numerous aspects such as multi-modal prediction (Li et al., 2017; Gupta et al., 2018), human social interactions (Alahi et al., 2016; Xu et al., 2022a) and scene context restrictions (Sadeghian et al., 2019; Liang et al., 2019), and have achieved remarkable progress. Beyond the above points, when reflecting on the sequential nature of trajectory prediction, *i.e.* an agent's presence in a scene is typically a long sequence and thus a series of consecutive predictions is performed over time, we believe here lies another essential information that is not well exploited in previous research all along, namely *individual feedback*.

Individual feedback refers to the information derived from the differences between the model's previous predictions and the ground-truth trajectory, and can provide valuable experience for subsequent predictions on the same agent. Particularly as illustrated in Fig. 1-a, when a model is continuously making predictions on a single agent, its previous predictions, *e.g.* those from several seconds ago, could already be verified by the agent's ground-truth trajectory which has become available through the progression of time. Such verification offers the individual feedback. Since the feedback includes references to the strengths and weaknesses of the model's predictions on the agent, if this information is properly utilized, it is able to heuristically guide to deliver more accurate predictions on this agent, and thereby brings overall improvements. However, none of the existing studies have paid much attention to exploring such information.

---

*Equal Contribution.
†Corresponding Author.

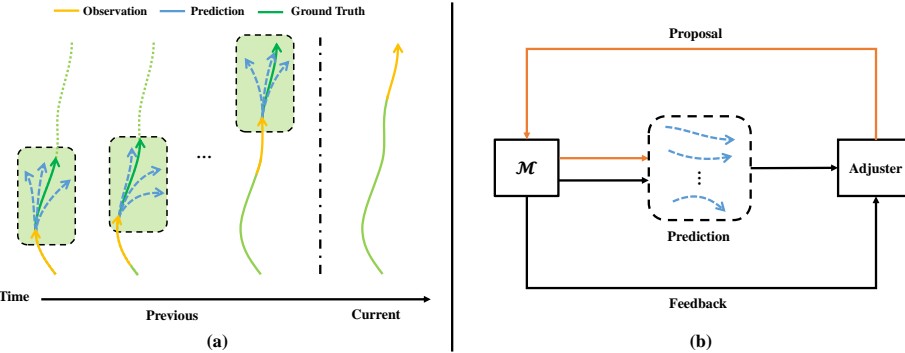

Figure 1: (a) Illustration of the individual feedback. Previous predictions of a target can be verified by ground truths through the progression of time, offering valuable experience on the strengths and weaknesses of the prediction model. (b) A brief schematic of the interactive adjustment process. $\mathcal{M}$ refers to the prediction model.

Still, it is non-trivial to effectively learn and leverage the feedback to adjust the original predictions into better ones. First, the feedback information cannot be simply integrated into the prediction model and learned end-to-end, since the calculation of individual feedback requires the output of the prediction model. Therefore, the feedback adjustment network should be developed as an external module to the prediction model. But this leads to a second problem as an external module cannot directly be aware of the restraints given by the prediction model when adjusting, *e.g.* the adjusted prediction may still violate social rules even though the prediction model has already learnt the social restrictions.

In this paper, we design the interactive adjustment network, *a.k.a.* IAN, to present a novel scheme on feedback modeling and utilization. IAN is an external module for the aforementioned first issue. As the trained prediction model is continuously performing predictions on a target individual, IAN first embeds the feedback from previous predictions and corresponding ground truths, and then aggregates all the feedback together to dynamically generate an adjuster specifically for the target. Thereby, the adjuster can adjust the model's current predictions into several proposals regarding future trajectories according to the integrated feedback information. Although these proposals can be directly decoded into trajectories as final predictions, this faces the aforementioned second issue. To this end, we generate the final predictions by querying the prediction model for trajectory candidates and further filtering candidates to figure out those with high confidence in candidate-proposal coherency as final results. In this way, the adjusted final predictions are not only optimized by the feedback but are also in line with the prediction model. A brief schematic of the whole interactive process is shown in Fig. 1-b, where the adjuster gets feedback information from the prediction model (black arrows) and the prediction model in turn leverages adjusted proposals from the adjuster to generate the final results for improved accuracy (orange arrows).

Considering the final results are provided by the prediction model rather than IAN itself, conventional loss functions for trajectory prediction such as L2 are not available since the gradient will not be back-propagated to IAN. To train this interactive architecture, we further raise a novel displacement expectation loss. Observing the confidence evaluated by the filter between a proposal and a candidate trajectory reveals the probability to select this candidate for the proposal, the expectation of displacement between the candidates and the ground-truth trajectory for a proposal indicates the error between the proposal and the ground truths. By optimizing with our proposed loss, IAN can be trained end-to-end to learn the feedback and leverage it to generate accurate proposals.

As an external module, IAN can be easily adopted to various prediction models. We conduct exhaustive experiments on three widely-used trajectory prediction benchmarks (Pellegrini et al., 2009; Leal-Taixé et al., 2014; Zhou et al., 2012; linouk23, 2016) with 6 representative prediction models (Gupta et al., 2018; Shi et al., 2021; Pang et al., 2021; Xu et al., 2022a; Shi et al., 2023; Bae et al., 2024), including state-of-the-art (Bae et al., 2024). The results demonstrate the great value of individual feedback, the superior effectiveness of IAN and the significant performance boost on trajectory prediction.

We summarize the contributions of our paper as follows.

- We study individual feedback, which reveals the prediction model's performance in predicting a specific agent. Individual feedback exists in the sequential nature of trajectory prediction tasks, and can be derived from the differences between a model's previous predictions and the ground-truth trajectory of the specific agent. To the best of our knowledge, we are the **first** to adopt such agent-specific information into trajectory prediction tasks.

- We analyze the properties of individual feedback and design an interactive adjustment network (IAN) to properly leverage individual feedback. The proposed IAN is a fully external module and can be easily adopted to other prediction models only with small computational overhead.

- Our experiments show that IAN significantly boosts the performances of multiple base prediction models on various datasets, proving the effectiveness of individual feedback.

## 2 RELATED WORKS

Human trajectory prediction (Hirakawa et al., 2018; Lee et al., 2017; Gupta et al., 2018; Shi et al., 2021; Sun et al., 2021; Xu et al., 2022a; Sun et al., 2022; Bae et al., 2022; Shi et al., 2023; Sun et al., 2023b;a; Dong et al., 2023; Li et al., 2024; Yang et al., 2024) is proposed to forecast the future movements of traffic agents given past observations. It has numerous important applications such as autonomous vehicles and robots (Hirakawa et al., 2018; Rudenko et al., 2019). Due to the fact that there is no single correct future, some works (Lee et al., 2017; Gupta et al., 2018) have paid their attention to multi-modal prediction, which aims at generating multiple possible future trajectories given a single observation to cover the uncertainty in the future. Based on the multi-modal setting, many recent works focus on exploiting various additional information apart from the target's past trajectory to aid in the prediction. For example, approaches like Sophie (Sadeghian et al., 2019), Trajectron++ (Salzmann et al., 2020) and SingularTrajectory (Bae et al., 2024) take scene context into consideration during prediction, while others like Social GAN (Gupta et al., 2018), SGCN (Shi et al., 2021), GroupNet (Xu et al., 2022a) and SocialCircle (Wong et al., 2024) design various methods to model the social interactions. Further, some recent works (Meng et al., 2022; Thakkar et al., 2024) tried to capture scene-specific patterns to adapt the predictions better fit the current scenario. However, there is another type of agent-specific information that has been neglected by previous works, namely individual feedback. It emerges from subsequent predictions of the model on the same agent, and offers experience from the past to improve the current prediction.

## 3 FORMULATION OF INDIVIDUAL FEEDBACK IN TRAJECTORY PREDICTION

In this section, we first briefly describe the task setting of trajectory prediction and then introduce the problem formulation of using individual feedback to improve the prediction performance. A list of key notations and their meanings are provided in Tab. 3 of the Technical Appendix for better reference.

**Trajectory Prediction** In the conventional setting of trajectory prediction, a prediction model $\mathcal{M}$ takes an observation sequence $O$ with length $\tau_{obs}$ of an agent as input and then predicts a series of trajectories $\hat{\mathbb{Y}}$ with length $\tau_{pred}$ for him. If the agent can be observed since timestep $T_a$, the model's prediction of this agent, $\hat{\mathbb{Y}}$, at timestep $T \geq \tau_{obs} + T_a - 1$ can be formulated as

$$\hat{\mathbb{Y}}_T = \mathcal{M}(O_T) = \left\{ \hat{Y}_T^i | i = 1, 2, \cdots, k \right\} \tag{1}$$

where $\hat{Y}_T^i$ is one of the predictions at timestep $T$ and $k$ is the number of required predictions. $Y_T$ is used to denote the agent's ground truth trajectory at timestep $T$. *For simplicity, we omit $T_a - 1$ by assuming $T_a = 1$ in the following context.*

**Individual Feedback** When $T \geq \tau_{obs} + \tau_{pred}$, the qualities of previous predictions $\{\hat{\mathbb{Y}}_{\tau_{obs}}, \ldots, \hat{\mathbb{Y}}_{T-\tau_{pred}}\}$ of an agent can be verified by the ground-truth trajectories

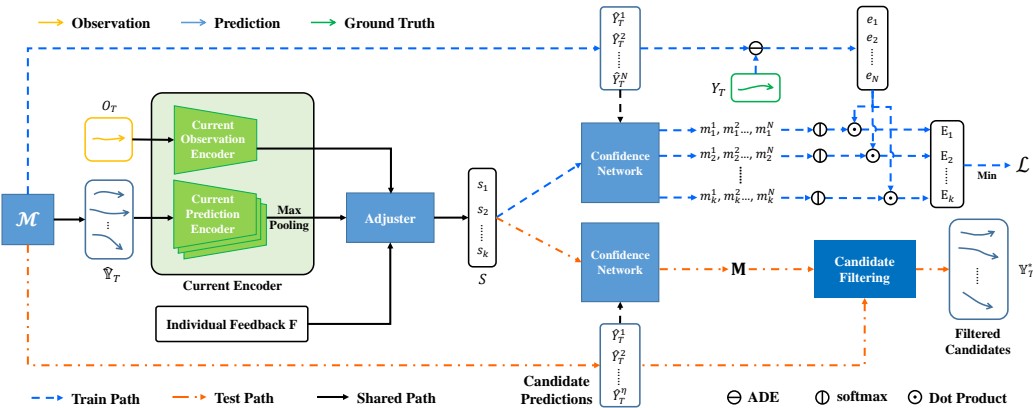

Figure 2: Illustration of the overall IAN structure including both training and testing stages. Train/test path refers to steps that are only present during training/testing. The 'Confidence Network's in the figure are in fact the same network, we duplicate its block representation for better clarity. **F** is acquired according to Eq. 3. The proposed displacement expectation loss is illustrated in the upper-right part of the figure.

$\{Y_{\tau_{obs}}, \ldots, Y_{T-\tau_{pred}}\}$[1]. We consider the similarity and disparity between the two bring individual feedback **F**, revealing the performance, including the strengths and weaknesses, of the model's predictions on this agent. This feedback information can be used to adjust the current predictions $\hat{\mathbb{Y}}_T$ to better results $\mathbb{Y}_T^*$ with a function $\mathcal{F}$, *i.e.* $\mathbb{Y}_T^* = \mathcal{F}(\hat{\mathbb{Y}}_T, \mathbf{F})$, **w.r.t.** $\mathbf{d}(\mathbb{Y}_T^*, Y_T) \leq \mathbf{d}(\hat{\mathbb{Y}}_T, Y_T)$, where **d** refers to the evaluation metrics for trajectory prediction.

## 4 INTERACTIVE ADJUSTMENT NETWORK

### 4.1 OVERVIEW

As motioned in Sec. 3, when $T \geq \tau_{obs} + \tau_{pred}$, we can learn pieces of feedback from previous predictions and corresponding ground truths with a feedback embedding module $f$ by

$$F_t = f\left(\hat{\mathbb{Y}}_t, Y_t\right), \tau_{obs} \leq t \leq T - \tau_{pred} \qquad (2)$$

Considering each piece of feedback information $F_t$ is related to the corresponding observed trajectory $O_t$, we raise a feedback aggregation operation $g$ that first integrates the observation feature into each piece of feedback and then aggregates all of them together as a whole. The full process can be written as

$$\mathbf{F} = G_F\left(\{(\hat{\mathbb{Y}}_t, Y_t, O_t)\}\right) = g\left(\{(F_t, O_t)\}\right), \tau_{obs} \leq t \leq T - \tau_{pred} \qquad (3)$$

where $G_F(\cdot)$ denotes the feedback generator and **F** is the individual feedback which is further used as a set of (dynamic) parameters of the adjuster $A_{\mathbf{F}}$.

In this manner, the adjuster is aware of individual feedback information, and can be used to adjust the current predictions $\hat{\mathbb{Y}}_T$ with the observation $O_T$ and generate a series of proposals $S$ regarding the future trajectories,

$$S = \{s_1, s_2, \cdots, s_k\} = A_{\mathbf{F}}\left(E_{curr}(O_T, \hat{\mathbb{Y}}_T)\right) \qquad (4)$$

where $E_{curr}$ refers to encoders for the current observation and predictions, and $k$ is the number of required predictions in the multi-modal setting. The proposals are then used to filter the candidates queried from the prediction model to get the final results

$$\mathbb{Y}_T^* = \Phi\left(\mathcal{M}, S\right) \qquad (5)$$

---

[1]$\{Y_{\tau_{obs}}, \ldots, Y_{T-\tau_{pred}}\}$ are ground-truth trajectories of previous timesteps, which can be already observed at the current time step T. We do NOT involve ground-truth trajectories of the current prediction $Y_T$ as input.

where $\Phi$ refers to the proposal query & filtering process.

Fig. 2 and Fig. 3 illustrate the architecture of our proposed approach. And in the following Sec. 4.2, Sec. 4.3 and Sec. 4.4, we respectively discuss the feedback generator, the adjuster and the query & filtering process in detail. Then in Sec. 4.5, we explain how to train IAN with a novel displacement expectation loss. The implementation details are provided in the technical appendices.

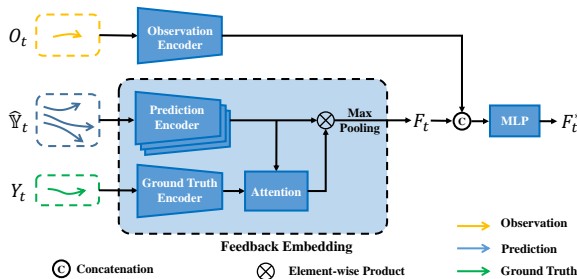

Figure 3: Illustration of feedback embedding process and observation integration. Here $\tau_{obs} \leq t \leq T - \tau_{pred}$.

### 4.2 FEEDBACK GENERATOR

According to Eq. 2 and 3, the feedback generator $G_F$ learns the whole individual feedback $\mathbf{F}$ for the current predictions $\hat{\mathbb{Y}}_T$ from a series of previous observations $\{O_t\}$, predictions $\{\hat{\mathbb{Y}}_t\}$, and corresponding ground truths $\{Y_t\}$ in $\tau_{obs} \leq t \leq T - \tau_{pred}$ with two modules, feedback embedding $f$ and feedback aggregation $g$.

**Feedback Embedding** To obtain each piece of feedback $F_t$, we first encode each trajectory $\hat{Y}_t^i \in \hat{\mathbb{Y}}_t$ and $Y_t$ into deep representations $\hat{R}_t^i$ and $R_t$ with a prediction encoder $E_{pred}$ and a ground-truth encoder $E_{gt}$

$$\hat{\mathbb{R}}_t = \left\{ \hat{R}_t^i = E_{pred}\left(\hat{Y}_t^i\right) | i = 1, 2, \cdots, k \right\}; \quad R_t = E_{gt}(Y_t) \tag{6}$$

Then, considering the attention mechanism is a common approach that helps models to learn the correlation between data, we introduce an attention gate to discovery the key similarity and disparity between features of the ground truth $R_t$ and each prediction $\hat{R}_t^i$

$$\delta_t^i = \text{softmax}\left(\frac{R_t \hat{R}_t^{i\top}}{\sqrt{d}}\right) \hat{R}_t^i \tag{7}$$

In this manner, the result $\delta_t^i$ indicates the verification of a prediction $\hat{Y}_t^i$ by the ground-truth $Y_t$. $d$ is the dimension of $R_t$. Finally, we aggregate $\{\delta_t^i | i = 1, 2, \cdots, k\}$ together to get a piece of feedback $F_t$ with max pooling, considering it is an effective and efficient symmetric operation.

$$F_t = \text{maxpool}\left(\{\delta_t^i\}\right) \tag{8}$$

**Feedback Aggregation** With pieces of feedback embedding $F_t$, we then aggregate them together for the whole individual feedback $\mathbf{F}$. Considering the prediction model predicts differently on distinct observations, the feedback will also vary among the observations. To this end, we first integrate the observation features into each feedback embedding,

$$F_t^* = \text{mlp}\left([E_{obs}(O_t), F_t]\right) \tag{9}$$

where $E_{obs}$ is the observation encoder, and $[\cdot, \cdot]$ denotes the concatenation operation. After that, all pieces of observation-aware feedback embedding $F_t^*$ are fed to an aggregation function for $\mathbf{F}$. We use max pooling in our implementation for its symmetric property, formally,

$$\mathbf{F} = \text{maxpool}\left(\{F_t^* | \tau_{obs} \leq t \leq T - \tau_{pred}\}\right) \tag{10}$$

### 4.3 ADJUSTER FOR PROPOSAL

In Eq. 4, an adjuster is derived from the individual feedback $\mathbf{F}$ and generates a series of proposals $S$ given the current observation $O_T$ and predictions $\hat{\mathbb{Y}}_T$. Inspired by previous works (Jia et al., 2016; Tian et al., 2020), we dynamically generate an mlp adjuster by regarding the feedback as its parameters, denoting as $A_{\mathbf{F}}$. That is $A_{\mathbf{F}}(\cdot) = \text{mlp}(\cdot; \mathbf{F})$. This mechanism enables the feedback information to directly influence the input of the adjuster, i.e. $\hat{\mathbb{Y}}_T$ and $O_T$, and offers $k$ adjusted proposals for future predictions as output. The encoders for $O_T$ and $\hat{\mathbb{Y}}_T$ share the same architecture with $E_{obs}$ and $E_{pred}$.

---

**Algorithm 1** Candidate Filtering

---

**Input**: Confidence Matrix $\mathbf{M} \in \mathbb{R}^{k \times \eta}$; Candidate Predictions $\{\hat{Y}_T^j | j = 1, 2, ..., \eta\}$;
**Output**: Set $\mathbb{Y}_T^*$ consisting of filtered candidates;
**Initialize**: $\mathbb{Y}_T^* = \{\}$; Index $p, q$;
   **while** $\mathbf{M} \neq \{-\infty\}_{k \times \eta}$ **do**
     $p, q = \arg\max_{p,q} \mathbf{M}$;
     Add $\hat{Y}_T^q$ to $\mathbb{Y}_T^*$;
     Set $\mathbf{M}_{p,:} = -\infty$, $\mathbf{M}_{:,q} = -\infty$;
   **end while**
   **return** $\mathbb{Y}_T^*$;

---

## 4.4 PROPOSAL QUERY & FILTERING

So far, we have adjusted the current predictions and acquired a set of proposals $S = \{s_i | i = 1, 2, ..., k\}$. Although these proposals can be directly decoded into exact trajectories as final results, these trajectories may contradict the prediction model since the adjustment process is not aware of the features learnt by the model. In this part, we introduce how to use such proposals to query the prediction model for prediction candidates and further filter them to get the qualified final results as Eq. 5.

Given the proposals $S$, we first query the prediction model $\mathcal{M}$ for $\eta$ candidate predictions $\{\hat{Y}_T^j | j = 1, 2, ..., \eta\}$ with $\eta \geq k$ (we use $\eta = 200$ in our experiments), then a confidence matrix can be calculated as

$$\mathbf{M} = [\mathbf{m}_i | i = 1, 2, \cdots, k] \in \mathbb{R}^{k \times \eta}, \ \mathbf{m}_i = \left[ \phi\left(\hat{Y}_T^j, s_i\right) | j = 1, 2, \cdots, \eta \right] \tag{11}$$

where $\phi(\cdot)$ is a confidence network, and $\mathbf{m}_i^j \in \mathbf{M}$ is the confidence of the $i$-th proposal being coherent with the $j$-th candidate prediction. We then run a greedy algorithm (shown in Alg. 1) to filter out $k$ candidates as the final results in $\mathbb{Y}_T^*$. As $k$ is usually a small value, *e.g.* 20, the computational overhead of this loop is negligible.

## 4.5 TRAINING

**Training Set Collection** We first distinguish the training/test set of the prediction model $\mathcal{M}$ and IAN with $\mathbb{P}$ and $\mathbb{A}$ respectively. A special point about $\mathbb{A}$ is that it requires not only the observations and the ground truths, but also the predictions from $\mathcal{M}$ according to Eq. 3. The former ones are available in $\mathbb{P}$ while the latter one is not. Therefore, these predictions should first be collected to build the training set of $\mathbb{A}$. Intuitively, they can be acquired by using $\mathcal{M}$ to infer on the training set of $\mathbb{P}$. Yet this practice is flawed. Since $\mathcal{M}$ has already been specifically optimized on the training set of $\mathbb{P}$ but has never seen the test set of $\mathbb{P}$, there is a considerable gap between the distribution of $\mathcal{M}$'s predictions on the training and test set of $\mathbb{P}$. This further leads to inconsistency between the distribution of training data and test data of $\mathbb{A}$.

To tackle this problem, we draw on the idea of $K$-fold cross validation. Specifically, the training set of $\mathbb{P}$ is first split into $K$ folds. By considering each fold of training data as a pseudo test set and the rest $K-1$ folds together as a new training set, $K$ sub-datasets are created. Then we train and test $\mathcal{M}$ on each sub-dataset, producing $K$ prediction models and $K$ groups of pseudo test set predictions. When $K$ is large enough, the prediction models trained on the sub-datasets will be similar as that trained on the full training set (*i.e.* the training set of $\mathbb{P}$), and thus the distribution of these predictions on the pseudo test sets will be consistent to that on the real test set produced by the model trained on the full training set. In this way, the $K$ groups of pseudo test set predictions can be reasonably used as the predictions for the training set of $\mathbb{A}$. We use $K = 5$ in our experiments.

**Displacement Expectation Loss** IAN cannot be directly optimized by the error of the predictions due to the fact that the predictions are not given by IAN and thus the gradient cannot be back propagated to IAN. To tackle this issue, we raise a novel displacement expectation loss for training and its application process is shown in Fig. 2.

During the training process of IAN, we first acquire $S$ with $k$ proposals for a training sample according to Eq. 4. Afterwards, we take $N$ predictions from the prediction model and calulate the confidence matrix $\mathbf{M} \in \mathbb{R}^{k \times N}$ following Eq. 11. We use $N = 200$ in our experiments. Meanwhile, we calculate the displacement error between all $N$ predictions and the ground-truth future trajectory $Y_T$

$$\mathbf{e} = \left[ e_j = d\left(\hat{Y}_T^j, Y_T\right) | j = 1, 2, \cdots, N\right] \in \mathbb{R}^N \tag{12}$$

where $d(\cdot)$ is the ADE distance.

Observing the confidence score reveals the probability to select a candidate for a certain proposal, with both the confidence score matrix $\mathbf{M}$ and the error vector $\mathbf{e}$, we calculate the expected error of each proposal as

$$\mathbb{E} = \{softmax(\mathbf{m}_i) \cdot \mathbf{e} | i = 1, 2, \cdots, k\} \tag{13}$$

which can reveal both the quality of the proposals and the effectiveness of the filter network. By optimizing the expectation error, IAN can gradually learn from the feedback to generate both reasonable proposals as well as a high-quality prediction-proposal filter.

Considering the multi-modal nature of trajectory prediction, simultaneously optimizing all the $k$ expectations will have a negative impact and cause mode collapse on the proposals, therefore, we refer to the winner-takes-all optimization technique and define our loss function as

$$\mathcal{L} = \min \mathbb{E} \tag{14}$$

## 5 EXPERIMENTS

### 5.1 EXPERIMENT SETTINGS

**Benchmarks**   We conduct experiments on the following three widely-used benchmarks. *ETH (Pellegrini et al., 2009)/UCY (Leal-Taixé et al., 2014) Dataset* is one of the most commonly used benchmarks. We follow Alahi et al. (2016) for the leave-one-out evaluation and observation/prediction horizon. *Grand Central Station Dataset (GCS) (Zhou et al., 2012)* contains trajectories extracted from a 30-min video recorded at the Grand Central Station. 8 steps (3.2 seconds) are for observation and 12 steps (4.8 seconds) are for prediction following Gupta et al. (2018). We split the first 80% of the dataset for training, and the rest 20% for test. *NBA Sports VU Dataset (NBA) (linouk23, 2016)* contains trajectories of all ten players in real NBA games. We adopt the traditional setting of 5 steps (2.0 seconds) for observation and 10 steps (4.0 seconds) for prediction. We select 50k samples in total from the 2015-2016 season with a split of 65%, 10%, 25% as training, validation and testing data following Li et al. (2020). It is worth noting that since we only consider data samples with at least one piece of feedback available and ignore the rest, *i.e.* those appear in the scene for less than $\tau_{obs} + \tau_{pred}$ timesteps, the reported baseline performances are different from their original value.

**Metrics**   We use the common metrics ADE/FDE for evaluation. In the multi-modal prediction setting, both metrics are calculated as the minimum over all $k$ trajectories. We set $k$ to 20 following the common setting (Gupta et al., 2018).

**Prediction Models**   We conduct experiments on the following five representative prediction models. *Social GAN (Gupta et al., 2018)*, a GAN based prediction framework modeling the social interactions with a pooling mechanism. *SGCN (Shi et al., 2021)*, a graph convolutional network that learns motion tendency with a temporal graph and social interactions with a directed spatial graph. *LB-EBM (Pang et al., 2021)*, a probabilistic model with cost function defined in the latent space to account for the movement history and social context. *GroupNet (Xu et al., 2022a)*, an encoding framework that models social interactions with multi-scale hypergraphs. We use GroupNet on CVAE with their official implementation (sjtuxcx, 2022). *TUTR (Shi et al., 2023)*, a transformer encoder-decoder architecture that unifies the trajectory prediction components, social interactions, and multimodal trajectory prediction. *SingularTrajectory (Bae et al., 2024)*, a diffusion-based universal trajectory prediction framework designed to bridge the performance gap across five tasks. We use official models for testing if available, otherwise we train models according to the official implementations.

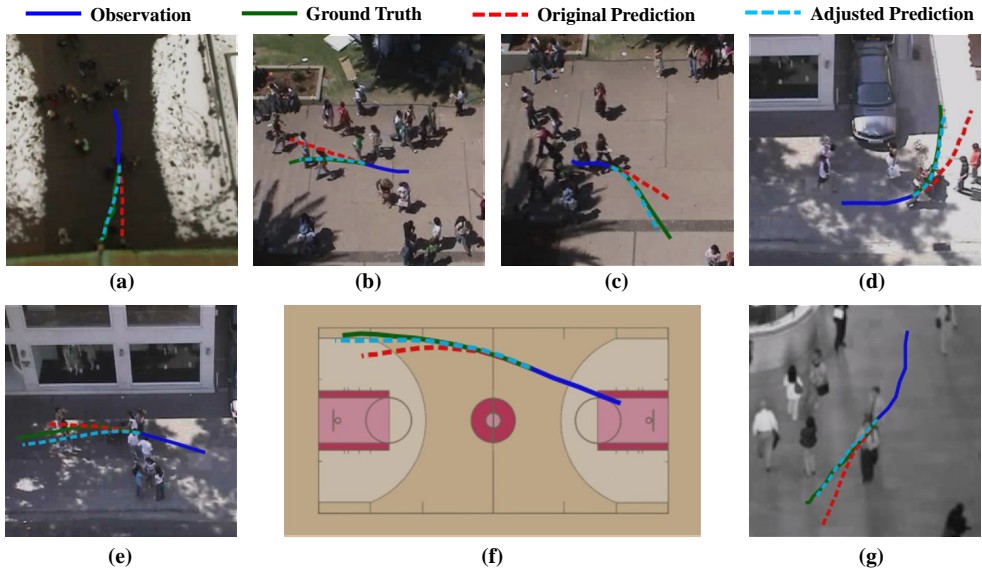

Figure 4: Qualitative analysis of the predicted trajectories before and after IAN is applied. The plotted predictions are the ones with minimum error among a total of 20 predictions.

Table 1: Performance (ADE/FDE) of representative prediction models before and after IAN is applied on three benchmarks (1:ETH/UCY; 2:GCS; 3:NBA). All the baseline performances are evaluated on models trained with corresponding official implementations. Since we ignore test samples with no feedback, the reported baseline results are different from the original values. ST stands for SingularTrajectory.

| ID | Method | Social GAN | SGCN | LB-EBM | GroupNet | TUTR | ST |
|---|---|---|---|---|---|---|---|
| 1 | Baseline | 0.22 / 0.39 | 0.28 / 0.48 | 0.20 / 0.36 | 0.21 / 0.35 | 0.23 / 0.39 | 0.20 / 0.30 |
| | w/ IAN | **0.20 / 0.34** | **0.26 / 0.44** | **0.18 / 0.31** | **0.19 / 0.30** | **0.22 / 0.36** | **0.19 / 0.29** |
| | *Impr.* | 9.1% / 12.8% | 7.1% / 8.3% | 10.0% / 13.9% | 9.5% / 14.3% | 4.3% / 7.7% | 5.0% / 3.3% |
| 2 | Baseline | 4.47 / 7.40 | 4.10 / 6.51 | 3.19 / 5.24 | 2.65 / 4.05 | 2.81 / 4.40 | 3.05 / 4.60 |
| | w/ IAN | **4.28 / 7.04** | **3.79 / 5.94** | **2.98 / 4.72** | **2.56 / 3.83** | **2.72 / 4.25** | **2.96 / 4.45** |
| | *Impr.* | 4.3% / 4.9% | 7.6% / 8.8% | 6.6% / 9.9% | 3.4% / 5.4% | 3.2% / 3.4% | 3.0% / 3.3% |
| 3 | Baseline | 1.53 / 2.24 | 1.56 / 2.46 | 1.40 / 2.08 | 1.17 / 1.64 | 1.23 / 1.93 | 1.24 / 1.60 |
| | w/ IAN | **1.47 / 2.11** | **1.44 / 2.19** | **1.34 / 1.94** | **1.13 / 1.56** | **1.19 / 1.84** | **1.20 / 1.55** |
| | *Impr.* | 3.9% / 5.8% | 7.7% / 11.0% | 4.3% / 6.7% | 3.4% / 4.9% | 3.3% / 4.7% | 3.2% / 3.1% |

## 5.2 MAIN RESULTS

We first quantitatively analyze the effectiveness of our approach. Results in Tab. 1 show that substantial performance improvements are achieved for all the prediction models on three benchmarks after applying IAN to model the individual feedback information. Particularly, improvements up to 10.0%/14.3% for ADE/FDE are achieved on strong baselines such as LB-EBM and GroupNet. In some cases, the improvements brought by IAN is about 3%. Such improvements are still significant on these benchmarks, as similar improvements were achieved by recent state-of-the-art prediction models (Shi et al., 2023; Bae et al., 2024) over their respective predecessors. We find such phenomenon intuitive as performances on these benchmarks advance towards the limits.

We further compare the adjusted predictions against the original ones with visualization in Fig. 4. These cases demonstrate that IAN can successfully adjust predictions to more accurate ones in terms of the moving direction (*e.g.* b, c), acuteness of turning (*e.g.* d), and velocity (*e.g.* e, f), *etc.*

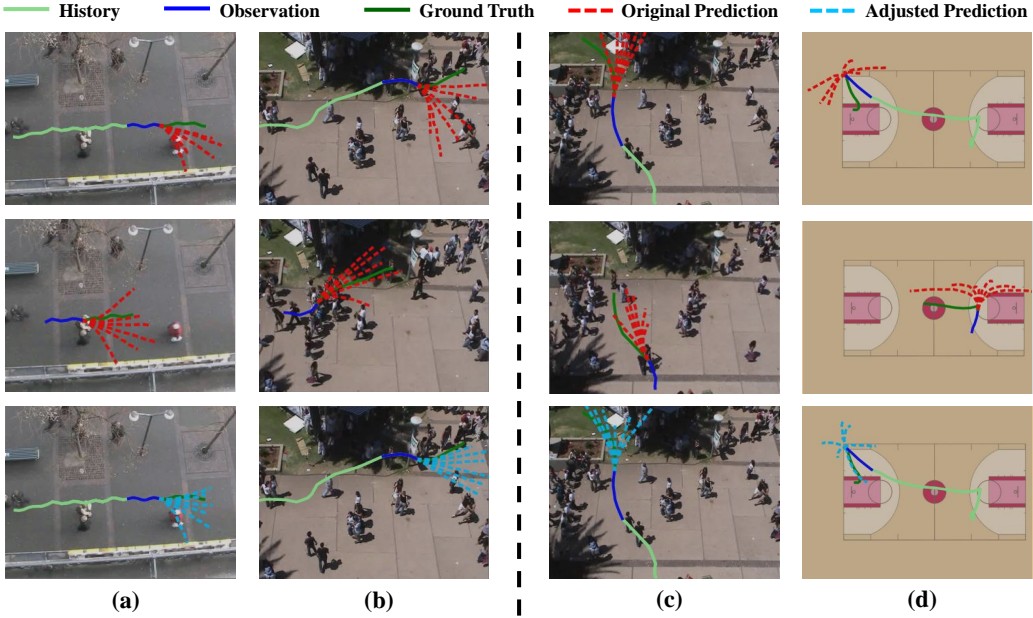

Figure 5: Examples of how IAN learns from feedback. (a, b): positive feedback, and (c, d): negative feedback. Figures in the first row are illustrations of current predictions before adjustment. The second row shows the corresponding feedback from some while ago. And the third row illustrates the predictions after adjustment.

## 5.3   ABLATION STUDY AND ANALYSIS

**Learning from Individual Feedback**   To better demonstrate how the individual feedback contributes to the adjustment, we show visualizations in Fig. 5 that IAN learns from either positive (a, b) or negative (c, d) feedback to adjust the current predictions into more accurate ones.

Fig. 5-a illustrates a jittery trajectory. The original predictions fail to focus on the true direction of the agent's movements and largely mislead by the jitters. Luckily, there is a piece of feedback from a similar situation that an accurate result is achieved with more diverse predictions. Based on the experience, IAN similarly increases the diversity of the current predictions and decreases the error. Fig. 5-b is another challenging case of double-turning. Leveraging the feedback of successfully predicting the other turning after the first one, IAN increases the tendency of predictions with opposite-direction turns in this similar scenario.

In Fig. 5-c and d, the feedback records a previous situation with similar observation and predictions as the current one, which finally turns out a failure according to the ground truth. Learning from this negative feedback, *i.e.* insufficient diversity in c and lack of sharp turning in d, IAN adjusts the original predictions to compensate the weaknesses of the original predictions and generates much better results.

**Contribution Analysis**   Compared with the baseline prediction models, our approach introduces additional historical information. To demonstrate that the improvement is induced by the feedback rather than additional information, we further compare with two ablative approaches which use the same historical information as ours. i) *Baseline w/ fullobs*: The baseline prediction model using all the observation since the predicted target appears instead of using the observation with a fixed length. ii) *IAN w/o feedback*: An IAN architecture where the individual feedback **F** is replaced by an embedding of the predicted target's full history trajectory, extracted by an LSTM. Results are shown in Tab. 2.

Although intuitively incorporating additional historical information would lead to performance improvements, simply adding additional observation directly to the baseline models (*i.e.* Baseline w/

Table 2: Comparison of ADE/FDE on different ablative approaches against IAN. ST stands for SingularTrajectory.

| $\mathcal{M}$ | Dataset | Baseline | Baseline w/ fullobs | IAN w/o feedback | Direct Decoding | IAN |
|---|---|---|---|---|---|---|
| LB-EBM | ETH/UCY | 0.20 / 0.36 | 0.31 / 0.56 | 0.19 / 0.33 | 0.23 / 0.38 | **0.18 / 0.31** |
| | GCS | 3.19 / 5.24 | 4.06 / 6.70 | 3.11 / 5.03 | 3.32 / 5.14 | **2.98 / 4.72** |
| | NBA | 1.40 / 2.08 | 1.78 / 2.72 | 1.36 / 2.00 | 1.44 / 2.01 | **1.34 / 1.94** |
| GroupNet | ETH/UCY | 0.21 / 0.35 | 0.27 / 0.49 | 0.20 / 0.33 | 0.21 / 0.33 | **0.19 / 0.30** |
| | GCS | 2.65 / 4.05 | 3.46 / 5.27 | 2.60 / 3.95 | 3.25 / 5.06 | **2.56 / 3.83** |
| | NBA | 1.17 / 1.64 | 1.49 / 1.92 | 1.16 / 1.59 | 1.44 / 2.00 | **1.13 / 1.56** |
| ST | ETH/UCY | 0.20 / 0.30 | 0.21 / 0.32 | 0.20 / 0.30 | 0.22 / 0.33 | **0.19 / 0.29** |
| | GCS | 3.05 / 4.60 | 3.15 / 4.71 | 3.03 / 4.56 | 3.37 / 5.16 | **2.96 / 4.45** |
| | NBA | 1.24 / 1.60 | 1.27 / 1.74 | 1.23 / 1.58 | 1.29 / 1.66 | **1.20 / 1.55** |

fullobs) actually resulted in performance decline. This indicates that the baseline networks do not successfully handle long observation with arbitrary length so that they may not exploit useful information but get additional noise. In comparison, the IAN architecture can effectively obtain a steady boost.

Compared with IAN w/o feedback, the original IAN achieves much better performance. This demonstrates that the modeling of feedback is effective to bring more improvements upon just using additional historical information.

**Interactive Architecture**  As is described in Sec. 4.4, our proposed IAN obtains the final results by querying the prediction model with proposals instead of directly decoding them. We argue that directly decoding the proposals into trajectories is flawed, since the adjustment process is not aware of the features learnt by the prediction model. In Tab. 2, we compare the results of direct decoding against IAN. While IAN achieves significant improvements, the direct decoding approach faces certain performance decrease on many of these experiments.

## 5.4  INFERENCE TIME

As an external module for a trajectory prediction model, IAN does not operate on the prediction model itself but rather after it. In other words, the prediction model and IAN are consecutive modules in a pipeline and can run simultaneously. Therefore, using IAN will not slow down the prediction model to produce predictions. Please refer to Sec. E.1 in the Technical Appendices for more details.

Under our test environments with a single RTX3090, IAN takes an average of 0.02 seconds to produce the adjusted predictions for an agent. As a reference, LB-EBM, GroupNet, TUTR and SingularTrajectory take 0.03, 0.04, 0.05 and 0.02 seconds respectively to make the predictions. Our approach can conduct inference at a high frequency while bringing substantial improvements.

## 6  CONCLUSION

In this paper, we study the individual feedback to reveal the prediction model's performance for a specific agent, which has not been studied in previous research. An interactive adjustment network (IAN) is then proposed to learn and leverage valuable experience from the past feedback to aid in the present prediction with small computational overhead. IAN analyzes and aggregates the feedback information from previous predictions on the target and uses it to make adjustments to current predictions in an interactive manner with the prediction model. Moreover, a novel displacement expectation loss is proposed to train the IAN. Exhaustive experiments demonstrate the effectiveness of our approach on multiple prediction models across various benchmarks.

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

## A NOTATIONS

As our approach involves more notations than average, we provide a table of key notations used in our paper and their meanings in Tab. 3 for better clarity.

Table 3: Notations and their respective meanings used in our paper.

| Notation | Meaning |
|---|---|
| $\tau_{obs}$ | The observation horizon, a constant value. |
| $\tau_{pred}$ | The prediction horizon, a constant value. |
| $T_a$ | The timestep when the prediction target appears (assumed equal to 1 for simplicity), *i.e.* target is observable from the start, in our paper. |
| $T$ | The timestep of 'current' prediction. |
| $\mathcal{M}$ | Prediction model, NOT IAN. |
| $O_*$ | Observation sequence with length $\tau_{obs}$, subscript denotes the timestep of its last element. |
| $Y_*$ | The target's ground truth future trajectory with length $\tau_{pred}$, subscript denotes the timestep right before its first element. |
| $\hat{Y}_*$ | The target's future trajectory predicted by $\mathcal{M}$, subscript denotes the timestep right before its first element. |
| $\hat{\mathbb{Y}}$ | A set containing $\hat{Y}$. |
| $k$ | The number of required predictions under multi-modal trajectory prediction setting. |
| $K$ | The number of splits used for training set collection, see Sec. 4.5. |

## B IAN PIPELINE

In order to better formulate the IAN's pipeline at both training and testing stages, we illustrate the pipeline via pseudo-codes in Alg. 2 and 3.

---

**Algorithm 2** IAN Training

**Input**: Training data $\mathbb{A}_{train} = \bigcup\limits_{a \in agents} \left\{ \left( \mathbb{H}_T^a, \hat{\mathbb{Y}}_T^a, Y_T^a, O_T^a \right) | \forall T \geq T_a + \tau_{obs} + \tau_{pred} - 1 \right\}$,

with $\mathbb{H}_T^a = \left\{ (\hat{\mathbb{Y}}_t^a, Y_t^a, O_t^a) | T_a + \tau_{obs} - 1 \leq t \leq T - \tau_{pred} \right\}$,

$\hat{\mathbb{Y}}_t^a = \mathcal{M}(O_t^a), \hat{\mathbb{Y}}_T^a = \mathcal{M}(O_T^a), |\hat{\mathbb{Y}}_t^a| = k, |\hat{\mathbb{Y}}_T^a| = N$;
**Output**: Trained feedback generator $G_F$, adjuster $A$, and confidence network $\phi$.
**Initialize**: $G_F, A, \phi$;

    **for all** $\left( \mathbb{H}_T^a, \hat{\mathbb{Y}}_T^a, Y_T^a, O_T^a \right) \in \mathbb{A}_{train}$ **do**
        Calculate $\mathbf{F} = G_F(\mathbb{H}_T^a)$ according to Eq. 6 - 10;
        Parameterize $A$ with $\mathbf{F}$ (Sec. 4.3): $A_{\mathbf{F}}(\cdot) = \text{mlp}(\cdot; \mathbf{F})$;
        Calculate $S = \{s_1, s_2, \cdots, s_k\}$ according to Eq. 4;
        Calculate confidence matrix $\mathbf{M}$ according to Eq. 11;
        Calculate displacement expectation loss $\mathcal{L}$ according to Eq. 12 - 14;
        Back-propagate and optimize $G_F, A, \phi$;
    **end for**
    **return** $G_F, A, \phi$;

---

## C IMPLEMENTATION DETAILS

In our implementation, we use LSTM networks for all the encoders in IAN, and the output sizes of these encoders are all 64. Each proposal generated by the adjuster has a dimension of 32. The adjuster is a four-layer mlp with embedding sizes of (128, 256, 512, 640, 640) and individual feedback $\mathbf{F}$ serves as the biases of its last two layers for low computational cost and a good convergence.

---

**Algorithm 3** IAN Inference

---

**Input**: $\left(\mathbb{H}_T^a, \hat{\mathbb{Y}}_T^a, O_T^a\right)$ with $T \geq T_a + \tau_{obs} + \tau_{pred} - 1$,

$\mathbb{H}_T^a = \left\{ (\hat{\mathbb{Y}}_t^a, Y_t^a, O_t^a) | T_a + \tau_{obs} - 1 \leq t \leq T - \tau_{pred} \right\}$,

$\hat{\mathbb{Y}}_t^a = \mathcal{M}(O_t^a), \hat{\mathbb{Y}}_T^a = \mathcal{M}(O_T^a), |\hat{\mathbb{Y}}_t^a| = k, |\hat{\mathbb{Y}}_T^a| = \eta$;

**Output**: Filtered trajectories $\mathbb{Y}_T^*$;

    Calculate $\mathbf{F} = G_F(\mathbb{H}_T^a)$ according to Eq. 6 - 10;
    Parameterize $A$ with $\mathbf{F}$ (Sec. 4.3): $A_{\mathbf{F}}(\cdot) = \text{mlp}(\cdot; \mathbf{F})$;
    Calculate $S = \{s_1, s_2, \cdots, s_k\}$ according to Eq. 4;
    Calculate confidence matrix $\mathbf{M}$ according to Eq. 11;
    **return** Candidate_Filtering($\mathbf{M}, \hat{\mathbb{Y}}_T^a$) (Alg. 1);

---

Accordingly, the mlp used during feedback aggregation has 3 layers and an output size of 1280. The confidence network first encodes the input trajectory into deep feature, then concatenate it with the proposal. The concatenated feature is fed to a triple-layer mlp with input size of 96 and outputs a single number as the confidence score. We use $\eta = N = 200$ for all prediction models except TUTR, where we set $\eta = N = L$ (notation $L$ indicates the number of 'general motion modes' in the TUTR paper) for ETH/UCY, and use $\eta = N = 80$ for the other datasets. During collection of the training set, the original training set is split into $K = 5$ folds. The network is trained for 30 epochs using Adam optimizer. In this paper, we primarily focus on proposing the idea of individual feedback and introducing how the IAN architecture can effectively leverage individual feedback to improve trajectory prediction. Therefore, we do not delve into adopting more complex structures for each module of IAN, *e.g.* using a heavier encoder instead of LSTM.

# D    ADDITIONAL RESULTS

## D.1    DETAILED PERFORMANCES

In Tab. 4, we give the detailed performances and widths of 95% confidence intervals of all the prediction models with/without IAN on all datasets. Improvements are achieved across all the sub-datasets, further proving IAN's effectiveness. The confidence intervals are acquired using results from 5 separate runs.

## D.2    ALTERNATE METRICS

To more comprehensively evaluate the performance of IAN, we additionally report the miss rates (Waymo, 2024) of base model predictions with and without IAN on ETH/UCY dataset in Tab. 5. We also report the brier-ADE/FDE following TUTR (Shi et al., 2023) in Tab. 6. These results demonstrate the consistent improvements provided by the IAN across various metrics.

## D.3    MORE ABLATIONS

**Number of Candidate Predictions**    In Sec. 4.4, we use the proposals to filter $\eta$ candidate predictions to get the final results. We investigate the influence of different values of $\eta$ in Tab. 7 (left). When $\eta$ is relatively small, the candidates are usually not sufficient to satisfy all the proposals with high confidence scores. Then as $\eta$ grows larger, the performance improves fast since more candidates that are highly coherent with the proposals are available. When $\eta$ is large enough, all proposals may have already been satisfied with high confidence scores and the performance changes little as $\eta$ varies.

**N in Displacement Expectation Loss**    In the proposed displacement expectation loss, the larger $N$ is, the closer the calculated expectation is to the ideal one, and thereby this tends to bring better performance. We show the influence of different values of $N$ in Tab. 7 (right).

Table 4: Detailed performances of IAN and widths of confidence intervals with various prediction models (1:ETH; 2:HOTEL; 3:UNIV; 4:ZARA1; 5:ZARA2; 6:GCS; 7:NBA). ST stands for SingularTrajectory.

|   | Method | SGAN | SGCN | LB-EBM | GroupNet | TUTR | ST |
|---|---|---|---|---|---|---|---|
| 1 | Baseline | 0.293 / 0.442 | 0.441 / 0.669 | 0.327 / 0.567 | 0.285 / 0.393 | 0.414 / 0.628 | 0.287 / 0.346 |
|   | $\epsilon$ | 0.006 / 0.009 | 0.010 / 0.013 | 0.007 / 0.010 | 0.003 / 0.007 | 0.008 / 0.011 | 0.003 / 0.006 |
|   | w/ IAN | **0.269 / 0.385** | **0.403 / 0.586** | **0.272 / 0.436** | **0.231 / 0.268** | **0.391 / 0.583** | **0.284 / 0.333** |
|   | $\epsilon$ | 0.005 / 0.007 | 0.008 / 0.013 | 0.007 / 0.011 | 0.004 / 0.007 | 0.009 / 0.011 | 0.003 / 0.006 |
| 2 | Baseline | 0.172 / 0.291 | 0.179 / 0.241 | 0.094 / 0.133 | 0.146 / 0.226 | 0.118 / 0.154 | 0.110 / 0.160 |
|   | $\epsilon$ | 0.001 / 0.002 | 0.002 / 0.002 | 0.001 / 0.001 | 0.001 / 0.002 | 0.001 / 0.001 | 0.001 / 0.002 |
|   | w/ IAN | **0.147 / 0.231** | **0.168 / 0.222** | **0.087 / 0.120** | **0.132 / 0.182** | **0.109 / 0.131** | **0.106 / 0.156** |
|   | $\epsilon$ | 0.001 / 0.002 | 0.001 / 0.001 | 0.001 / 0.001 | 0.001 / 0.001 | 0.001 / 0.002 | 0.001 / 0.001 |
| 3 | Baseline | 0.275 / 0.511 | 0.338 / 0.602 | 0.258 / 0.497 | 0.259 / 0.497 | 0.271 / 0.472 | 0.273 / 0.463 |
|   | $\epsilon$ | 0.004 / 0.005 | 0.004 / 0.007 | 0.002 / 0.004 | 0.003 / 0.004 | 0.003 / 0.005 | 0.002 / 0.004 |
|   | w/ IAN | **0.266 / 0.493** | **0.314 / 0.548** | **0.230 / 0.424** | **0.235 / 0.438** | **0.258 / 0.461** | **0.261 / 0.457** |
|   | $\epsilon$ | 0.003 / 0.004 | 0.002 / 0.004 | 0.002 / 0.002 | 0.002 / 0.003 | 0.003 / 0.004 | 0.002 / 0.003 |
| 4 | Baseline | 0.211 / 0.394 | 0.290 / 0.547 | 0.198 / 0.385 | 0.201 / 0.374 | 0.182 / 0.341 | 0.187 / 0.337 |
|   | $\epsilon$ | 0.002 / 0.003 | 0.002 / 0.003 | 0.001 / 0.002 | 0.001 / 0.003 | 0.002 / 0.002 | 0.001 / 0.002 |
|   | w/ IAN | **0.200 / 0.364** | **0.281 / 0.519** | **0.187 / 0.358** | **0.200 / 0.363** | **0.178 / 0.331** | **0.183 / 0.327** |
|   | $\epsilon$ | 0.002 / 0.003 | 0.002 / 0.002 | 0.001 / 0.002 | 0.001 / 0.003 | 0.001 / 0.002 | 0.001 / 0.003 |
| 5 | Baseline | 0.152 / 0.301 | 0.165 / 0.320 | 0.106 / 0.211 | 0.136 / 0.273 | 0.162 / 0.343 | 0.119 / 0.197 |
|   | $\epsilon$ | 0.001 / 0.002 | 0.002 / 0.002 | 0.001 / 0.001 | 0.001 / 0.002 | 0.001 / 0.002 | 0.001 / 0.002 |
|   | w/ IAN | **0.131 / 0.243** | **0.159 / 0.303** | **0.100 / 0.191** | **0.126 / 0.242** | **0.151 / 0.320** | **0.110 / 0.192** |
|   | $\epsilon$ | 0.001 / 0.002 | 0.001 / 0.002 | 0.002 / 0.002 | 0.001 / 0.002 | 0.001 / 0.001 | 0.001 / 0.002 |
| 6 | Baseline | 4.47 / 7.40 | 4.10 / 6.51 | 3.19 / 5.24 | 2.65 / 4.05 | 2.81 / 4.40 | 3.05 / 4.60 |
|   | $\epsilon$ | 0.05 / 0.08 | 0.05 / 0.07 | 0.06 / 0.11 | 0.04 / 0.08 | 0.03 / 0.06 | 0.03 / 0.07 |
|   | w/ IAN | **4.28 / 7.04** | **3.79 / 5.94** | **2.98 / 4.72** | **2.56 / 3.83** | **2.72 / 4.25** | **2.96 / 4.45** |
|   | $\epsilon$ | 0.05 / 0.07 | 0.05 / 0.08 | 0.05 / 0.06 | 0.03 / 0.07 | 0.04 / 0.05 | 0.03 / 0.06 |
| 7 | Baseline | 1.53 / 2.24 | 1.56 / 2.46 | 1.40 / 2.08 | 1.17 / 1.64 | 1.23 / 1.93 | 1.24 / 1.60 |
|   | $\epsilon$ | 0.01 / 0.02 | 0.01 / 0.02 | 0.01 / 0.02 | 0.01 / 0.03 | 0.02 / 0.03 | 0.01 / 0.02 |
|   | w/ IAN | **1.47 / 2.11** | **1.44 / 2.19** | **1.34 / 1.94** | **1.13 / 1.56** | **1.19 / 1.84** | **1.20 / 1.55** |
|   | $\epsilon$ | 0.01 / 0.02 | 0.01 / 0.03 | 0.01 / 0.02 | 0.01 / 0.02 | 0.01 / 0.02 | 0.01 / 0.02 |

Table 5: Performance (miss rate) of prediction models before and after IAN is applied. ST stands for SingularTrajectory.

| $\mathcal{M}$ | Method | ETH | HOTEL | UNIV | ZARA1 | ZARA2 |
|---|---|---|---|---|---|---|
| Social-GAN | Baseline | 0.0909 | 0.0289 | 0.0611 | 0.0236 | 0.0505 |
|   | w/ IAN | **0.0909** | **0.0120** | **0.0473** | **0.0124** | **0.0372** |
| SGCN | Baseline | 0.1212 | 0.0216 | 0.1249 | 0.0933 | 0.0844 |
|   | w/ IAN | **0.0909** | **0.0144** | **0.0993** | **0.0833** | **0.0755** |
| LB-EBM | Baseline | 0.1320 | 0.0062 | 0.0594 | 0.0185 | 0.0350 |
|   | w/ IAN | **0.0644** | **0.0000** | **0.0405** | **0.0116** | **0.0216** |
| GroupNet | Baseline | 0.0303 | 0.0072 | 0.0609 | 0.0087 | 0.0263 |
|   | w/ IAN | **0.0303** | **0.0000** | **0.0391** | **0.0062** | **0.0193** |
| TUTR | Baseline | 0.0595 | 0.0040 | 0.0452 | 0.0417 | 0.0260 |
|   | w/ IAN | **0.0536** | **0.0040** | **0.0386** | **0.0243** | **0.0242** |
| ST | Baseline | 0.0606 | 0.0000 | 0.0241 | 0.0050 | 0.0133 |
|   | w/ IAN | **0.0606** | **0.0000** | **0.0231** | **0.0037** | **0.0128** |

Table 6: Performance (brier-ADE/FDE) of TUTR before and after IAN is applied.

| $\mathcal{M}$ | Method | ETH | HOTEL | UNIV | ZARA1 | ZARA2 |
|---|---|---|---|---|---|---|
| TUTR | Baseline | 1.18 / 1.39 | 0.64 / 0.68 | 0.98 / 1.19 | 1.01 / 1.16 | 0.73 / 0.91 |
|   | w/ IAN | **1.14 / 1.33** | **0.63 / 0.66** | **0.96 / 1.16** | **0.99 / 1.15** | **0.72 / 0.89** |

Table 7: [Left] Comparison between different maximum number of candidates $\eta$ on ETH/UCY. [Right] Comparison between different $N$ on ETH/UCY. ST stands for SingularTrajectory.

| $\mathcal{M}$ | | $\eta$ | 50 | 100 | 200 | $N$ | 50 | 100 | 200 |
|---|---|---|---|---|---|---|---|---|---|
| LB-EBM | ADE | | 0.183 | 0.176 | **0.175** | ADE | 0.183 | 0.181 | **0.175** |
| | FDE | | 0.324 | 0.308 | **0.306** | FDE | 0.325 | 0.320 | **0.306** |
| GroupNet | ADE | | 0.201 | 0.193 | **0.192** | ADE | 0.200 | 0.196 | **0.192** |
| | FDE | | 0.312 | 0.296 | **0.296** | FDE | 0.324 | 0.313 | **0.296** |
| ST | ADE | | 0.192 | 0.190 | **0.189** | ADE | 0.192 | 0.190 | **0.189** |
| | FDE | | 0.298 | 0.294 | **0.293** | FDE | 0.299 | 0.295 | **0.293** |

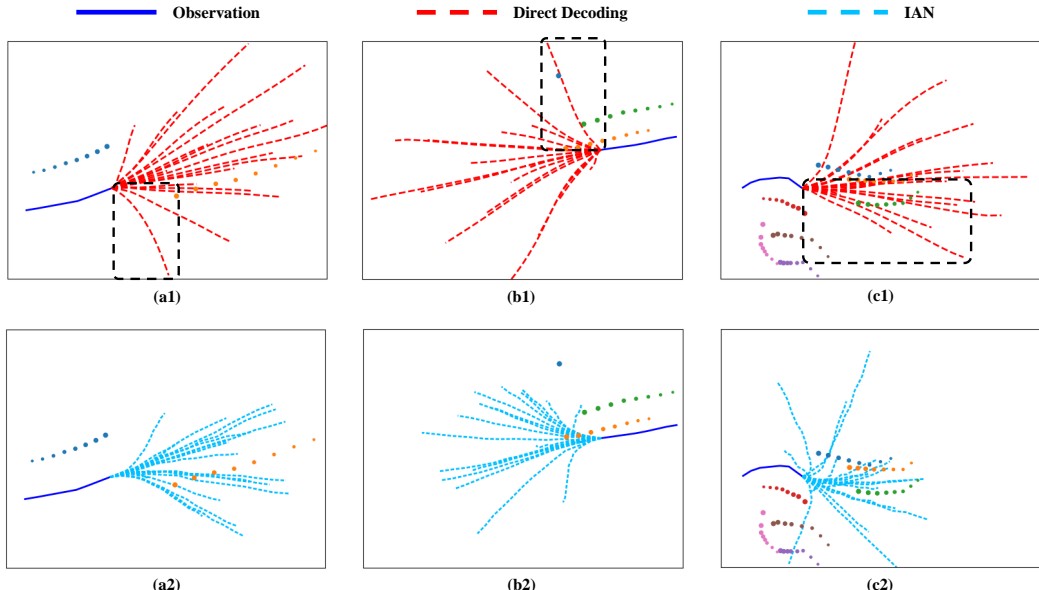

Figure 6: Comparisons of predictions acquired by direct proposal decoding and IAN. The observed trajectory is mark with solid lines whereas predictions are marked with dashed ones. The social agents are marked with dots with various colors, and the direction that the dots get bigger indicate the moving direction of an agent.

## D.4 IAN *v.s.* DIRECT PROPOSAL DECODING

To qualitatively demonstrate the weakness of direct proposal decoding, *i.e.* the predictions of direct decoding cannot follow the features learnt by the prediction model and thereby they may be flawed, we compare the predictions generated by direct decoding against those acquired by querying in Fig. 6. We use a social-aware approach GroupNet as the prediction model.

For each of the three cases, all of the 20 predictions are plotted. It is clear that predictions given by direct proposal decoding are not aware of the social behaviors and face a serious problem of social impossibility (marked with black rectangles), although the prediction model has already learnt the social features. For example, in Fig. 6-a1, the prediction at the bottom is highly probable to collide with the orange agent. Similarly, the top prediction in Fig. 6-b1 moves directly through an neighboring agent who stands still. A more sophisticated example in Fig. 6-c shows that the predictions from direct proposal decoding exhibit little consideration to these agents, leading to numerous predictions with a high chance of collision. In comparison, this issue is very well addressed by adopting our approach. Particularly in Fig. 6-c2, our results show clear tendencies of avoiding the green oncoming agents.

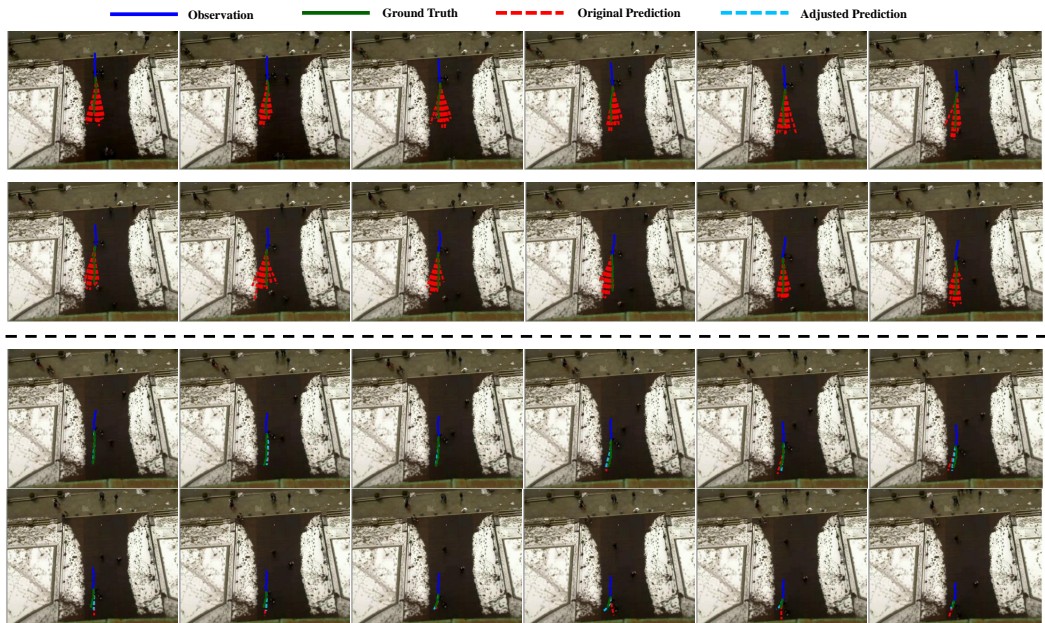

Figure 7: Consecutive visualizations of feedback pieces and adjustments on the same agent over time. [**Top**] In each of the first 12 frames we show the predictions $\hat{\mathbb{Y}}_t$ (red), ground truth $Y_t$ (green), observation $O_t$ (blue), such information will be used in the future by IAN to adjust future predictions. Note that during this period of time, IAN have no access to any feedback information and does not take effect. [**Bottom**] In frames 13-24 we show IAN continuously improving the prediction accuracy by adjusting. Note that IAN at frame $i(13 \leq i \leq 24)$ only have access to feedback information from frames $j_i \in \{j|1 \leq j \leq i - 12\}$, and the ground truth for frame $i$ is not leaked to the IAN.

## D.5    MORE QUALITATIVE RESULTS

In Fig. 7, we we give 24 consecutive frames of predictions on the same agent. In the first $\tau_{pred} = 12$ frames we show the observation (dark blue), predicted trajectories (red), as well as the ground truth (green). IAN does not take effect during these 12 frames since no ground truth trajectories are available during this period. Note that the predicted trajectories are constantly longer than the ground truth during these 12 frames, indicating that the predictions tend to have a higher speed.

From the 13th frame, we show the observation (dark blue), best prediction before adjustment (red), best prediction after adjustment (light blue), and the ground truth (green). Note that the $i$-th frame $(13 \leq i \leq 24)$ only has ground truths from frame $j_i$ $(1 \leq j_i \leq i - \tau_{pred})$ available for calculating the individual feedback $\mathbf{F}$. These frames with IAN operational clearly show that the adjusted predictions tend to be slower than those prior to adjustments, indicating IAN's capability of acquiring information from the individual feedback *i.e.* predictions tends to be faster than the ground truth. Further, the adjusted trajectory also show trends of gradual improvement.

## D.6    DISPLACEMENT EXPECTATION LOSS: ANALYSIS

In this paper, we introduced a novel displacement expectation loss to train the IAN. The loss is devised based on the winner-takes-all optimization technique, which is widely used multi-modal trajectory prediction studies (Chai et al., 2019; Xu et al., 2022a; Shi et al., 2023). Further, the value of the displacement expectation (Eq. 13) is bounded by the minimum and maximum displacement errors of the predictions from the base prediction model. Therefore, ideally, the displacement expectation will converge to the minimum displacement error during optimization. We additionally provide example training curves in Fig. 8 to demonstrate its converging property.

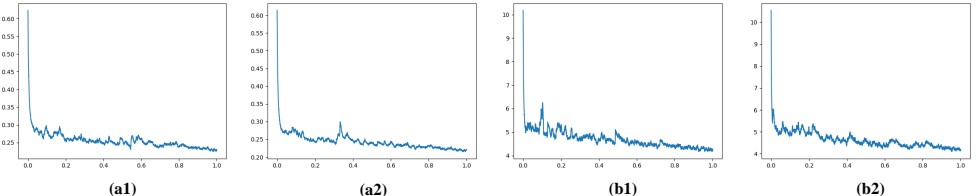

Figure 8: Example training curves of displacement expectation loss from our experiments. (a1&a2) Training of IAN under GroupNet + ZARA1 repeated twice. (b1&b2) Training of IAN under LB-EBM + GCS repeated twice.

# E MORE DISCUSSIONS

## E.1 INFERENCE TIME

In Sec 5.4 of our paper, we discussed about the base prediction model $\mathcal{M}$ and IAN running simultaneously in a pipeline as consecutive models. Here we provide an illustration of such pipeline in Fig. 9.

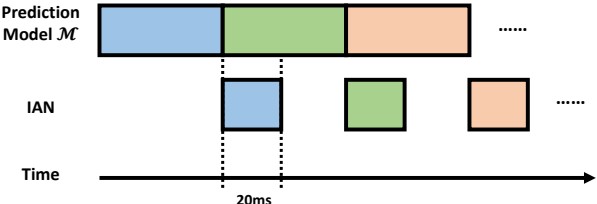

Figure 9: Illustration of the prediction model $\mathcal{M}$ and IAN as a pipeline. Each prediction task is identified with a unique color. IAN and $\mathcal{M}$ can run at the same time without interfering each other, and thereby using IAN will not slow down $\mathcal{M}$ to produce new predictions. The additional runtime for getting better predictions with IAN will not accumulate and is always 20ms.

## E.2 DIFFERENCE WITH TEACHER-ENFORCING

As teacher-enforcing technique also involves the ground truth, here we clarify that our approach is essentially different from that, to avoid misunderstanding.

Teacher-enforcing is a training technique that feeds ground truths back into the RNN after each step during the training phase. In comparison, 1) our approach is **an external module beyond the base prediction model** (teacher-enforcing is **a training technique for the base model**) and 2) our approach works by **adjusting the predictions of the base model in the test phase** (teacher-enforcing works only in the **training phase**). Therefore, our approach is essentially different from teacher-enforcing.

## E.3 DIFFERENCE WITH PRIOR ADAPTATION WORKS

Prior to IAN, there have been other researches that use the idea of adaptation for better results (Xu et al., 2022b; Ivanovic et al., 2023), which involves using data from the test domain/scene to train/fine-tune the prediction model. In comparison, data from the test domain/scene is not used to train IAN. Another major difference between IAN and these approaches is that IAN uses agent-specific information, *i.e.* individual feedback, and adjusts the predictions for each agent respectively. On the other hand, typical adaptation approaches aggregate the trajectories of all the agents in the

test domain/scene as a whole and use it to modify the predictions for all the agents in the test domain/scene.

### E.4 DIFFERENCE WITH TEST-TIME TRAINING/ONLINE LEARNING

While IAN operates by adjusting the prediction model's outputs, none of the network (both the base prediction model and the IAN) parameters are updated during test-time. Therefore, IAN is entirely different from test-time training/fine-tuning approaches. In addition, test-time training is in fact not compatible with our aim, *i.e.* individual feedback, due to its individual-specific nature. Such trait indicates that when actually deployed, a distinct model for each of the individuals present in the scene needs to be stored and updated, which can be extremely costly for online deployment on embodied agents such as autonomous vehicles and robots.

### E.5 LIMITATION AND FUTURE WORKS

One scenario where predictions are adjusted in a worse way occurs when an agent whose past trajectory is smooth and almost straight suddenly changes direction. Before the change in direction, the base model performs better with more concentrated predictions (*e.g.* a pattern that all predictions are near straight lines with minor changes in direction). Under such circumstances, the adjuster may notice this pattern and therefore adjusts scattered predictions to concentrate more, which will harm the prediction accuracy when the change in direction occurs, examples are shown in Fig. 10. The figure shows some examples of this case. We are actively addressing this issue as our future work.

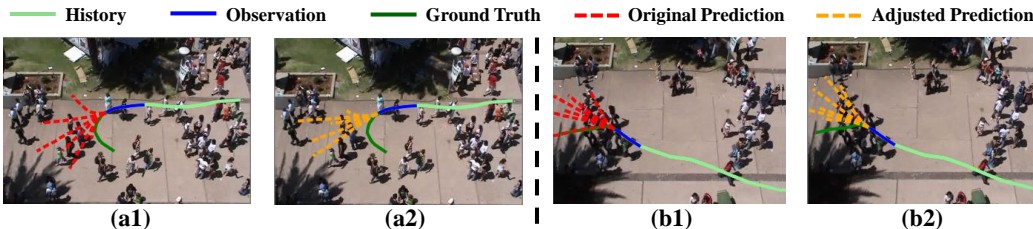

Figure 10: Examples of the adjuster worsening predictions.

