# OpenReview forum: "Interactive Adjustment for Human Trajectory Prediction with Individual Feedback"
_ICLR.cc/2025/Conference — ICLR 2025 Poster_

### Official Review · Reviewer_Bpfi · 2024-11-02

**Soundness:** 3
**Presentation:** 3
**Contribution:** 3
**Rating:** 8
**Confidence:** 5

**Summary:**

This paper introduces the concept of individual feedback in trajectory prediction problems by designing an Interaction Adjustment Network (IAN). The IAN, comprising a feedback generator, an adjuster, and a filtering process, is designed to be an external module that can be seamlessly integrated with other trajectory prediction models. To ensure its adaptability, the authors have designed a displacement loss function to train the IAN. Experiments have shown the IAN's efficacy on widely adopted benchmarks for trajectory prediction.

**Strengths:**

Following are the strengths of this paper:
1. The introduction of individual feedback to enhance trajectory prediction by integrating agent-specific past trajectories with groundtruth is promising.
2. Evaluation of the proposal IAN network on trajectory prediction benchmarks highlights their claim.
3. As an external module, IAN can be integrated with various trajectory prediction methods.
4. The presentation of results is good.

**Weaknesses:**

However, this paper provides a promising way to improve the trajectory prediction method by incorporating the agent-specific past trajectories and groundtruths, here are some comments that need to be addressed
1. Since the authors introduced the individual feedback in the paper and provided the intuition, it would be good to see any theoretical justification for why it is important.
2. It would be good to see if the authors can address that incorporating the agent-specific previous trajectories may not lead to overfitting the base-model prediction.
3. Another important concern is that, since the IAN is heavily dependent on the base-model prediction if there are accumulated errors in the base-model prediction, then the proposal generated from the IAN would also be spurious. What kind of risk assessment should be incorporated into the IAN?
4. It would be good to see if there is more explanation on the architecture of IAN, including the feature generation network, what kind of encoder architecture is used, and the rationale behind it.
5. Another thing that would be good is for the authors to include at least some description of the process happening in the figures to help the reader understand the figures.
6. In the conclusion, the authors should include the limitations of their work and possible future directions. It would also be good to see if authors can report the failure cases.

**Questions:**

However, this paper provides a promising way to improve the trajectory prediction method by incorporating the agent-specific past trajectories and groundtruths, here are some comments that need to be addressed
1. Since the authors introduced the individual feedback in the paper and provided the intuition, it would be good to see any theoretical justification for why it is important.
2. It would be good to see if the authors can address that incorporating the agent-specific previous trajectories may not lead to overfitting the base-model prediction.
3. Another important concern is that, since the IAN is heavily dependent on the base-model prediction if there are accumulated errors in the base-model prediction, then the proposal generated from the IAN would also be spurious. What kind of risk assessment should be incorporated into the IAN?
4. It would be good to see if there is more explanation on the architecture of IAN, including the feature generation network, what kind of encoder architecture is used, and the rationale behind it.
5. Another thing that would be good is for the authors to include at least some description of the process happening in the figures to help the reader understand the figures.
6. In the conclusion, the authors should include the limitations of their work and possible future directions. It would also be good to see if authors can report the failure cases.

---

> ### Author Response · Authors · 2024-11-24
> **Responses to Reviewer Bpfi**
>
> We deeply appreciate the reviewer for the positive feedback and the thoughtful questions, and are glad to provide answers below. We have also revised the paper according to the suggestions. We have uploaded the revised version of our submission and highlighted changes in the main paper. The original version (P21-36) is attached below the Technical Appendices of the revised version as reference.
>
> >Q1: Theoretical justification.
>
> We are glad to provide a theoretical justification as follows. We consider the base prediction model $\mathcal{M}$ learns a distribution $P _\mathcal{M}$ and makes predictions for **a specific novel agent $\mathcal{A}$** at time step $T$ according to $P _\mathcal{M}$ given the observation of agent $\mathcal{A}$. Since $\mathcal{M}$ is optimized on a full set of training samples involving numerous individuals **other than $\mathcal{A}$**, there could be a gap between its learned distribution $P _\mathcal{M}$ and the real future trajectory distribution $P _\mathcal{R}$ of the agent $\mathcal{A}$. Therefore, we introduce individual feedback $\mathbf{F}$ of the agent $\mathcal{A}$, and our approach IAN as a function $f$, to produce $P _\mathcal{A} = f(P _\mathcal{M}, \mathbf{F})$ as a better approximation to the real distribution $P _\mathcal{R}$.
>
> >Q2: Overfitting the base model prediction.
>
> We thank the reviewer for raising this thoughtful question. In this paper we use individual feedback to adjust base model's predictions so that they are more consistent to the target agent's specific patterns. During such process, the base model's parameters are not updated and therefore has no risk of overfitting.
>
> >Q3: Errors in base model prediction.
>
> We would like to highlight that in IAN, the proposals are generated based on **both** base model predictions and individual feedback $\mathbf{F}$. It is our intention that the adjuster **uses the individual feedback to** generate proposals aiming at correcting potential errors of the base model predictions and therefore improve the performance.
>
> We deeply appreciate the reviewer's suggestion on incorporating risk assessment measures. By assuming that risk assessment means "evaluating and managing uncertainties and potential hazards that could arise from predicted movements of agents", we suggest that one possible aspect of risk assessment is to check the predictions against collision with other agents.
>
> >Q4: More Explanations on IAN's architecture.
>
> We thank the reviewer for the suggestion and kindly remind that such discussions are already provided in Sec.B of the Technical Appendices in the original version.
>
> >Q5: Description of figures.
>
> We appreciate the reviewer for raising the concern and have added more descriptions of the process in the captions.
>
> >Q6: Limitation and future works.
>
> Thank you for the suggestion! We have added discussion on limitations, failure cases and future works in Sec.E.5 of the Technical Appendices in the revised version.

---

> > ### Comment · Reviewer_Bpfi · 2024-11-26
> >
> > Thank you for providing the response. All my questions are answered!

---

> ### Author Response · Authors · 2024-12-04
> **Thank You for Your Efforts in Reviewing Our Submission**
>
> Dear Reviewer Bpfi,
>
> As the discussion period has come to an end, we would like to express our gratitude for the time you devoted to reviewing our paper. We deeply appreciate your thought-provoking questions, which have been very helpful in improving our submission. We also take great pride in having effectively answered them.
>
> Best regards,
>
> Authors of Submission 7210

---

### Official Review · Reviewer_ncs6 · 2024-11-02

**Soundness:** 3
**Presentation:** 2
**Contribution:** 2
**Rating:** 6
**Confidence:** 3

**Summary:**

In this paper, an individual feedback framework (termed IAN) for Trajectory Prediction is proposed that dynamically adjusts the confidence score conditioned on prediction consistency between prediction and GT trajectories. Through addtional feedback generator during training, a consistency score could be derived through attention in supervising the prediction condifence. Comprehensive experiment results demonstrate the effectiveness of IAN module when collaborating with a series of state-of-the-art predictors.

**Strengths:**

1. Novel design for trajectory prediction pipeline considering closed-loop decoding (or individual feedback). Through proposed feedback generater and adjuster model in IAN, the multi-modal prediction garners extra feedback consistency supervisions in prediction confidence.

2. Comprehensive experimental evaluations. 1) Broad improvements are reported across various datasets when integrating IAN with several SOTA motion prediction frameworks; 2) Clear comparison with other decoding strategies.

**Weaknesses:**

1. Unclear writings in methodology: It is partly clear to understand the feedback mechanism by Figure 2, However, by simply go through the content the reviewer could hardly understand the feedback generation process between training and testing. Hence, it is better to provide additional Algorithm part for clearer understanding.

2. Additional evaluation for the feedback. In Tab1-2, minADE/minFDE is a direct metric in mearusing the best-case similarity for multi-modal predicted trajectory. However, it is unclear whether the confidence are well-performed by the feedback enhancement. Hence, several extra results measured by Brier-minADE/FDE, MissRate, or mAP seems needed.

**Questions:**

1. How is the feedback being connected (with A_F for example) in Eq3-4 and Eq9-10, and being differentiated between training and testing stage?

---

> ### Author Response · Authors · 2024-11-24
> **Responses to Reviewer ncs6**
>
> We thank the reviewer for the positive feedback as well as suggestions for improvements, and gladly provide responses below. We have uploaded the revised version of our submission and highlighted changes in the main paper. The original version (P21-36) is attached below the Technical Appendices of the revised version as reference.
>
> >W1: Unclear writings.
>
> We thank the reviewer for the suggestion and have added the algorithm parts in Sec.B of the Technical Appendices in the revised version for clearer understanding.
>
> >W2: Additional evaluation.
>
> Thank you for the suggestion and we have added MissRate and Brier-ADE/FDE into the evaluation metrics. The results are shown in Tab.5 and Tab.6 in the Technical Appendices of the revised version. Here we only evaluate Brier-ADE/FDE using TUTR as base prediction model since only TUTR (among all base prediction models used in our paper) is capable of outputting confidence with its predictions.
>
> >Q1: Connecting the feedback.
>
> As stated in L265-267 and L677-678 of our original version, we connect individual feedback, *i.e.* $\mathbf{F}$, into the networks by using it as the parameters of the adjuster. Such approach is naturally differentiable in both training and testing stages.

---

> > ### Comment · Reviewer_ncs6 · 2024-11-27
> >
> > Dear authors,
> >
> > Thanks for your response. I would like to maintain my current scoring.

---

> > > ### Author Response · Authors · 2024-12-04
> > > **Thank You for Your Efforts in Reviewing Our Submission**
> > >
> > > Dear Reviewer ncs6,
> > >
> > > As the discussion period has come to an end, we would like to express our gratitude for the time you devoted to reviewing our paper. We are particularly grateful for your suggestions on improvements for better understanding and additional metrics, and we are also pleased to see your acknowledgment of our response.
> > >
> > > Best regards,
> > >
> > > Authors of Submission 7210

---

### Official Review · Reviewer_dXFb · 2024-11-02

**Soundness:** 3
**Presentation:** 2
**Contribution:** 2
**Rating:** 6
**Confidence:** 4

**Summary:**

The paper proposes an Interactive Adjustment Network (IAN) that leverages individual feedback from previous predictions to improve human trajectory prediction models. The idea is to use the differences between earlier predictions and actual trajectories (ground truths) to adjust future predictions for the same agent. The authors claim that their method can be applied as an external module to various existing prediction models and that it significantly boosts performance on datasets.

**Strengths:**

- The concept of utilizing individual feedback from previous predictions is interesting.
- The proposed method is designed to be model-agnostic and can be applied to various trajectory prediction frameworks.
- Experiments are conducted on multiple datasets and with several baseline models.

**Weaknesses:**

Major issues:
- The paper's biggest assumption is the fact is that IAN relies on having immediate access to the ground truth trajectories of agents to compute feedback. In real-world applications, example autonomous driving or robotics, such ground truth data is usually never available in real-time.
- The concept of using feedback from previous predictions is not entirely new. Adaptive models and online learning techniques have long incorporated past errors to improve future predictions. The bigger issue is the lack of adequate differentiation of IAN from existing methods i.e., a thorough literature review that situates its contributions within the broader context.
- While the authors introduced a new loss function, there is no analysis of the properties of the displacement expectation loss or proofs of convergence and stability.
- There is a lot of poorly defined notation, making it difficult to understand the proposed method fully. It would make sense to move the table 3 from the appendix (which has the notation!) to the main paper.

There are some issues with the presentation.

Writing:
- Section 3, Trajectory Prediction (just before equation 1): \hat{Y} of "him"
- Section 4.5, "At the beginning of this paragraph"
- Section 5.4, Last sentence: Our approach can "inference" at a high frequency
- Personal preference: please use "Figure X represents ..." (not abbreviation) instead of "Fig. X represents..." because your captions have the former.

Images:
- The image labels for Figure 2 are too small to read -- had to Zoom 150% to read it
- Figure 2, it is hard to tell the difference between Train Path and Test Path, the arrows look very similar, please consider a different color.
- Figure 4, 5, it is hard to see lighter colors.

**Questions:**

1. How do you justify the assumption that ground truth trajectories are available in real-time for computing feedback? In what practical scenarios would this be feasible?
2. How does your method differ from existing online learning approaches that adapt models based on prediction errors?
3. How does the computational cost of IAN affect inference time in real-world applications (even a small delay in prediction can go a long way)?
4. Are there experiments to isolate the effects of each component of your model? If so, please provide detailed results.
Statistical Significance: Are the reported improvements statistically significant? Can you include variance measures or confidence intervals?
5. How does your model handle agents for whom no feedback data is available?

---

> ### Author Response · Authors · 2024-11-24
> **Responses to Reviewer dXFb**
>
> We thank the reviewer for the comments and especially the detailed suggestions to improve our presentation. In the responses below we address to the reviewer's concerns in detail and welcome any further discussions. We have uploaded the revised version of our submission and highlighted changes in the main paper. The original version (P21-36) is attached below the Technical Appendices of the revised version as reference.
>
> >W1 & Q1: Access to ground truths of previous predictions.
>
> We believe that **as long as an agent appears continuously in the scene for a duration longer than $\tau_{obs} + \tau_{pred}$**, it is feasible **in all practical scenarios** for the ground-truth trajectories of the agent's **previous** predictions to be available for computing the agent's individual feedback. Here, $\tau_{obs}$ denotes the observation horizon and $\tau_{pred}$ denotes the prediction horizon.
>
> We justify this point by clarifying that, according to the formulation of individual feedback in **L156-183** of the original version, the ground-truth trajectories of **previous** predictions have already become **the observation trajectories at the current timestep**.
>
> - Specifically, assuming an agent appears at time step 1 and $T$ denotes the current timestep, the ground-truth trajectories of "previous predictions made from time step $\tau_{obs}$ to $T-\tau_{pred}$" are actually the observation trajectory of the agent from time step $\tau_{obs} + 1$ to $T$.
>
> - For example, the ground-truth trajectory of previous prediction made at time step $T-\tau_{pred}$ is the observation trajectory from time step $T-\tau_{pred}+1$ to $T$.
>
> Therefore, the ground truths of previous predictions are **naturally available** through the progression of time in practical scenarios.
>
>
> >W2 & Q2: Differences from online learning approaches.
>
> The essential difference between IAN and online learning approaches is that IAN **does not update its parameters** at test stage, as opposed to the very essential feature of online learning or test-time training approaches. This point fundamentally distinguishes IAN and online learning approaches, making them inherently different.
>
> We have provided discussions about our differences with online learning/test-time training approaches and why the online learning approaches are not suitable for agent-specific adjustments in Section D.2 of the original version, and have further extended the discussions to adaptive works in Section E.3 of the Technical Appendices in the revised version.
>
> >W3: Analysis of loss function.
>
> We thank the reviewer for the reminder and have added loss analysis in Section D.6 of the revised version.
>
> >W4: Moving Table 3 to the main paper.
>
> We thank the reviewer for this suggestion! We have added a reference to Table 3 at the beginning of Section 3 to help address the notation issues, since we find it challenging to move Table 3 to the main paper due to the page limitations.
>
> >W5: Presentation.
>
> We greatly appreciate the detailed suggestions provided in the comments and have improved these points in our revised version.
>
> >Q3: The inference time of IAN.
>
> The average inference time of IAN is 0.02 seconds (L525). Assuming that the speed of a human is 1.5 m/s on average, he/she can travel only about 3 cm in such a short time (0.02s). Therefore, we suggest that the additional inference time is negligible.
>
> >Q4-1: Experiments on component isolation.
>
> Currently our IAN features three main components, namely the feedback generator, the adjuster and the confidence network. Among these components, both the adjuster and the confidence network are essential to the pipeline and removing them would make IAN inoperable. Therefore, only the feedback generator can be removed from the pipeline as an ablation study, which we have provided in Table 2 (column "IAN w/o feedback").
>
> >Q4-2: Confidence Intervals.
>
> We thank the reviewer for the suggestion, and have added confidence intervals of the main experiments in Table 4 of the Technical Appendices in the revised version.
>
> >Q5: For agents no feedback data is available.
>
> If individual feedback is unavailable for an agent (meaning that the agent appears continuously in the scene for a duration *shorter* than $\tau_{obs} + \tau_{pred}$), IAN will not be applied to the agent. In such cases, the outputs of the prediction model will be used directly as the final outputs.

---

> > ### Comment · Reviewer_dXFb · 2024-11-25
> > **Response to author (reviewer: dXFb)**
> >
> > Dear author(s), thank you very much for the efforts put into answering my questions/clarifying the weakness. After going through the response to my comment (and to the comments of the other reviewers) and the updated version of the paper, I have decided to change my score to 6 (marginally above the acceptance threshold).

---

> > > ### Author Response · Authors · 2024-12-04
> > > **Thank You for Your Efforts in Reviewing Our Submission**
> > >
> > > Dear Reviewer dXFb,
> > >
> > > As the discussion period has come to an end, we would like to express our gratitude for the time you devoted to reviewing our paper. We are particularly grateful for your detailed suggestions to improve our presentation as they require a thorough reading of the entire paper, and are more than glad to see your recognition of our responses. Your increase in the rating means a lot to us.
> > >
> > > Best regards,
> > >
> > > Authors of Submission 7210

---

### Official Review · Reviewer_ykSs · 2024-11-03

**Soundness:** 3
**Presentation:** 3
**Contribution:** 1
**Rating:** 3
**Confidence:** 5

**Summary:**

In this work, an online learning methodology is proposed for trajectory prediction based on the insight that ground truth observations checking past predictions (named feedback) can be collected as time goes on. To leverage this insight, the authors propose an interactive adjustment network (IAN) and evaluate its efficacy using a variety of trajectory prediction algorithms on a set of pedestrian and athlete motion datasets.

**Strengths:**

The idea is sound and sensible.

The writing is clear and it is easy to follow the core idea being proposed.

There are a wide variety of baseline approaches used to demonstrate the core idea.

**Weaknesses:**

In contrast to the claims in the paper, this work is not the first to study feedback in trajectory predictions. Accordingly, the primary weakness of this work is a lack of discussion and comparisons to prior adaptive prediction work that investigates this same idea. Notable examples (which focus on both temporal adaptation, equivalent to individual feedback, and geographic adaptation) include:
* Y. Xu, L. Wang, Y. Wang, and Y. Fu, "Adaptive trajectory prediction
via transferable GNN," in IEEE Conf. on Computer Vision and Pattern
Recognition, 2022.
* B. Ivanovic, J. Harrison, and M. Pavone, "Expanding the deployment envelope of behavior prediction via adaptive meta-learning," in IEEE Conf. on Robotics and Automation, 2023.

The experimental setup relies primarily on small-scale pedestrian-only datasets. Further, these datasets are quite old (ETH/UCY are 10-15 years old, GCS is more than 10 years old, the NBA dataset is almost 10 years old), meaning performance has largely saturated on them and it is difficult to tell if the proposed method significantly improves upon baselines.

As stated in Section 5.4, a prediction model and the IAN are _consecutive_ modules, meaning they have to run one after the other (and not simultaneously as stated in Line 523). This means that there is a 20ms overall runtime increase as a result of adding the IAN.

**Questions:**

The most important question: How does this approach compare to prior works on adaptive trajectory prediction? Why is IAN better or more preferable to use by practitioners?

How does this approach perform on modern large-scale datasets with multiple types of agents and varying dynamics, e.g., Waymo Open Motion Dataset, nuPlan? These datasets also have less saturated metrics which should make it easier for the IAN's performance improvements to stand out.

---

> ### Author Response · Authors · 2024-11-24
> **Responses to Reviewer ykSs (Part 1)**
>
> We deeply appreciate the reviewer's effort on reviewing our submission. Yet we respectfully suggest that there may have some misconceptions regarding our paper. We explain in detail below and are happy to answer any further questions. We have uploaded the revised version of our submission and highlighted changes in the main paper. The original version (P21-36) is attached below the Technical Appendices of the revised version as reference.
>
> >W1 & Q1: IAN *v.s.* prior works on adaptive trajectory prediction
>
> We respectfully refute the comments since we have found **some erroneous statements in the comments** of this weakness and question, which may stem from some misunderstandings by the reviewer regarding our paper. Therefore, we believe **it is unjustified to reject our submission based on this point**.
>
> We will address the comments systematically in the following order and rectify the inaccuracies within.
>
> 1. The comments say "this work is not the first to study feedback in trajectory predictions", however, we actually write "we are the first to adopt **such agent-specific information** into trajectory prediction tasks" in L112 of the original version.
>
>     Here, *such agent-specific information* refers to "**individual** feedback, which reveals the prediction model’s performance in predicting **a specific agent** (an agent means a pedestrian or athlete)" in L108 of the original version. Please refer to the paragraph beginning from L155 of the original version for the mathematical formulation of individual feedback.
>
>     Having conducted a thorough and comprehensive literature review, we are confident that our claim of being the first to study individual feedback (**as distinct from other types of feedback or adaptation**) is not overstated.
>
> 2. The statement "prior adaptive prediction work [1,2] that investigates *this same idea*" and "[1,2] focus on *both temporal adaptation, equivalent to individual feedback*, and geographic adaptation" is inaccurate.
>
>     As suggested in the first point, we refute this statement by stating that our proposed individual feedback is very different from the temporal adaptation used in [1,2].
>
>     By clarifying basic concepts that "a training/test domain" as "a specific scene with multiple moving agents" to avoid confusion, we explain how [1,2] and ours operate
>
>     - [1,2]: The temporal adaptation actually uses the past trajectories of **all the agents in the test scene as a whole** to **train or fine-tune** the prediction model and **update its parameters**. Then, the prediction model trained or fine-tuned with the past trajectories of **all the agents in the test scene** is applied to predict the future trajectories of **all the agents**.
>
>     - Ours: We collect individual feedback of each **specific (single)** agent and use IAN to adjust the **output** of the prediction model (**not** fine-tune the prediction model and update its parameter) for each agent **respectively**, *i.e.* individual feedback of a specific agent is only used to adjust the predictions on itself.
>
>     Therefore, there are two significant differences between ours and [1,2]
>
>     - Individual feedback is **agent-specific**, which differs from the **scene-specific** adaptation in [1,2].
>
>     - [1,2] use data from the test domain to train or fine-tune the prediction model. In comparison, we **only** use the data from the train domain to train IAN as a separate model and do not use any data from the test domain for training. *We also discussed about why our approach is different with test-time fine-tuning and why test-time fine-tuning is not suitable for agent-specific adjustments in Sec.D.2 (L806) of the original version.*
>
> Finally, we draw the following conclusions and answer the questions in the comments based on the responses above.
>
> 1. We suggest that our **agent-specific** individual feedback is a novel idea, different from previous work that studies **scene-specific** adaptations. Further, we do not use data from the test domain to train the IAN, which is different from prior adaptive works such as [1,2] that use data from the test domain to train the prediction model.
>
> 2. *Why is IAN better or more preferable to use by practitioners?*
>
>     When we adopt approaches like [1,2], which are originally designed for scene-specific adaptation, for agent-specific adjustments, we need to train many separate prediction models with their own parameters for every agent in the scene. This is extremely costly for online deployment on autonomous vehicles and robots.
>
>     In comparison, we propose IAN as an innovative solution to agent-specific adjustment, which brings little computational overhead.
>
> 3. *How does this approach compare to prior works on adaptive trajectory prediction?*
>
>     As mentioned above, prior works like [1,2] are originally designed for scene-specific adaptation and not suitable for agent-specific adjustment, we suggest that this comparison is unnecessary and of limited significance.

---

> > ### Author Response · Authors · 2024-11-24
> > **Responses to Reviewer ykSs (Part 2)**
> >
> > >W2 & Q2: Datasets for evaluation.
> >
> > We respectfully argue that we should **not** be blamed for using datasets such as ETH/UCY for evaluation, since they have been widely adopted as benchmarks in many latest papers [3-13] in top conferences for **human** trajectory prediction, including those used as baseline prediction models in our paper, *e.g.* SingularTrajectory and TUTR. Hence, it is imperative to evaluate on these same benchmarks for fair comparisons.
> >
> > In addition, as the performances on the benchmark has saturated, performance improvements tend to be more difficult to achieve. Therefore, we also suggest that the improvements on these datasets clearly demonstrate the effectiveness of our method.
> >
> > We greatly appreciate the suggestion to evaluate on modern large-scale datasets. We are currently working on implementing our approach on MTR++ using Waymo. We will provide such results as soon as they are available.
> >
> > Nevertheless, we also respectfully suggest that our paper, as is stated in the title, focuses on **human** trajectory prediction. Therefore, as is done in [1] (used by the reviewer as reference) as well as many similar latest studies [3-13], it is justified to not use datasets like Waymo Open Motion Dataset, which includes non-human agents such as vehicles, for evaluation. Therefore, we should not be penalized for not evaluating on such datasets.
> >
> > >W3: 20ms overall runtime increase.
> >
> > We thank the reviewer for pointing this out. We suggest that a typical human would only travel about 3cm (at 1.5m/s) within 20ms, and therefore we consider the additional runtime a negligible delay.
> >
> > We also make a clarification of the term "simultaneously" in our revised version. Please refer to Sec.E.1 of the revised version.
> >
> > ### **References**
> >
> > 1. Y. Xu, L. Wang, Y. Wang, and Y. Fu, "Adaptive trajectory prediction via transferable GNN," 2022 IEEE/CVF Conference on Computer Vision and Pattern Recognition (CVPR).
> > 2. B. Ivanovic, J. Harrison, and M. Pavone, "Expanding the deployment envelope of behavior prediction via adaptive meta-learning," 2023 IEEE Conf. on Robotics and Automation.
> > 3. I. Bae, Y. -J. Park and H. -G. Jeon, "SingularTrajectory: Universal Trajectory Predictor Using Diffusion Model," 2024 IEEE/CVF Conference on Computer Vision and Pattern Recognition (CVPR).
> > 4. C. Wong, B. Xia, Z. Zou, Y. Wang and X. You, "SocialCircle: Learning the Angle-based Social Interaction Representation for Pedestrian Trajectory Prediction," 2024 IEEE/CVF Conference on Computer Vision and Pattern Recognition (CVPR).
> > 5. Lin, X., Liang, T., Lai, J., and Hu, J.-F., "Progressive Pretext Task Learning for Human Trajectory Prediction," 2024 European Conference on Computer Vision (ECCV).
> > 6. I. Bae, J. Lee and H. -G. Jeon. "Can Language Beat Numerical Regression? Language-Based Multimodal Trajectory Prediction," 2024 IEEE/CVF Conference on Computer Vision and Pattern Recognition (CVPR)
> > 7. S. Kim, H. -g. Chi, H. Lim, K. Ramani, J. Kim and S. Kim, "Higher-order Relational Reasoning for Pedestrian Trajectory Prediction," 2024 IEEE/CVF Conference on Computer Vision and Pattern Recognition (CVPR).
> > 8. Lee, S., Lee, J., Yu, Y., Kim, T., and Lee, K., "MART: MultiscAle Relational Transformer Networks for Multi-agent Trajectory Prediction," 2024 European Conference on Computer Vision (ECCV).
> > 9. Y. Su, Y. Li, W. Wang, J. Zhou, and X. Li, "A Unified Environmental Network for Pedestrian Trajectory Prediction," AAAI 2024.
> > 10. W. Xiang, H. YIN, H. Wang, and X. Jin, "SocialCVAE: Predicting Pedestrian Trajectory via Interaction Conditioned Latents," AAAI 2024.
> > 11. E. Weng, H. Hoshino, D. Ramanan and K. Kitani, "Joint Metrics Matter: A Better Standard for Trajectory Forecasting," 2023 IEEE/CVF International Conference on Computer Vision (ICCV).
> > 12. Y. Dong, L. Wang, S. Zhou and G. Hua, "Sparse Instance Conditioned Multimodal Trajectory Prediction," 2023 IEEE/CVF International Conference on Computer Vision (ICCV).
> > 13. I. Bae, J. Oh and H. -G. Jeon, "EigenTrajectory: Low-Rank Descriptors for Multi-Modal Trajectory Forecasting," 2023 IEEE/CVF International Conference on Computer Vision (ICCV).

---

> > > ### Author Response · Authors · 2024-12-04
> > > **Updates on the Waymo experiments**
> > >
> > > We provide our evaluation on the Waymo Open Dataset, **while emphasizing that we should not be criticized for not evaluating on such datasets as discussed in our previous responses**.
> > >
> > > |  | minADE / minFDE |
> > > |:--:|:--:|
> > > |Baseline|0.6341 / 1.2595|
> > > |w/ IAN|0.6227 / 1.2379|
> > > |*Impr.*|1.8% / 1.7% |
> > >
> > > To conduct the experiment on the Waymo Open Dataset, we had to create our own data split since the official validation and test splits do not support the evaluation of IAN. We explain in detail below.
> > >
> > > - IAN requires the feedback from previous predictions on a specific agent to adjust current predictions for the same agent. However, given the Waymo motion prediction setting (*i.e.* 1s observation and 8s prediction), each data sample in the validation set is trimmed to exactly 9 seconds long (1s + 8s) whereas those in test set only contain one-second observations. It is clear that neither the validation nor the test set supports the acquisition of individual feedback, therefore, we cannot evaluate IAN on them.
> > >
> > > - Unlike the validation and test sets, the official training set of the Waymo Dataset contains 20-second segments, allowing agents within these segments to appear long enough for individual feedback to be available.
> > >
> > > - Based on the explanations above, we split the official training set (1000 `training_20s` records) of the Waymo Dataset to create our own training (first 800 records) and test (last 200 records) sets to conduct this experiment. We use the new training set to train both the baseline network and IAN, and evaluate on the new test set, resulting in the results shown above.
> > >
> > > The experiment shows that our proposed IAN achieved a 1.8% / 1.7% improvement over the baseline. We suggest that this is a substantial improvement on the Waymo dataset. For reference, MTR++[1] (TPAMI 2024) achieved a 0.46% / 0.96% improvement over the previous SOTA HDGT[2] (TPAMI 2023) on Waymo's test set.
> > >
> > > ### **References**
> > > 1. S. Shi, L. Jiang, D. Dai and B. Schiele, "MTR++: Multi-Agent Motion Prediction with Symmetric Scene Modeling and Guided Intention Querying," IEEE Transactions on Pattern Analysis and Machine Intelligence, 2024.
> > > 2. X. Jia, P. Wu, L. Chen, Y. Liu, H. Li and J. Yan. "HDGT: Heterogeneous Driving Graph Transformer for Multi-Agent Trajectory Prediction via Scene Encoding," IEEE Transactions on Pattern Analysis and Machine Intelligence, 2023.

---

> ### Comment · Area_Chair_CMSG · 2024-12-01
>
> Dear Reviewer,
>
> Could you please look at authors feedback and see if your concerns (esp. the novelty / first work / comparison to prior literature) are addressed?
>
> Thanks,
> AC

---

> ### Comment · Reviewer_ykSs · 2024-12-02
> **Response to author feedback**
>
> I thank the authors for their thorough response. To dive deeper into the two significant differences between the proposed approach and [1,2] (focusing in on the comparison to [2]):
>
> 1. Individual feedback is agent-specific, which differs from the scene-specific adaptation in [1,2].
>
> Agent-specific feedback is a special case of the update equation (3) in the correction step of [2] ($w_{t+1|t+1} = w_{t+1|t} + K_{t+1}e_{t+1}$ is agent-specific when $K_{t+1}$ is diagonal)
>
> 2. [1,2] use data from the test domain to train or fine-tune the prediction model. In comparison, we only use the data from the train domain to train IAN as a separate model and do not use any data from the test domain for training. We also discussed about why our approach is different with test-time fine-tuning and why test-time fine-tuning is not suitable for agent-specific adjustments in Sec.D.2 (L806) of the original version.
>
> The original writing (and author feedback) states that "Such trait indicates that when actually deployed, a distinct model for each of the individuals present in the scene needs to be stored and updated, which can be extremely costly for online deployment on embodied agents such as autonomous vehicles and robots." However, one of the core points of [2] is that, rather than updating the weights of the entire neural network (which indeed would be extremely costly), one only needs to update the last layer of the neural network (which entails a single prediction and correction step as detailed in Equations (2) and (3) of [2], which are akin to the prediction and correction step of a Kalman filter, which are extremely efficient to execute and are already executed, for example, as part of onboard robotic state estimation algorithms). To state this directly: _there is no backpropagation happening onboard in [2]!_
>
> For these reasons, I am maintaining my original review score.

---

> > ### Author Response · Authors · 2024-12-03
> > **Responses to Reviewer ykSs**
> >
> > We thank the reviewer for sharing his/her views on our posts. However, we humbly suggest that it is unjustified to question the novelty of our submission based on such viewpoints.
> >
> > > 1: Agent-specific feedback is a special case of the update equation (3) in the correction step of [2] when $K_{t+1}$ is diagonal.
> >
> > After carefully studying [2], we have discovered that the reviewer's viewpoint, *i.e.* "$w_{t+1|t+1} = w_{t+1|t} + K_{t+1}e_{t+1}$ is agent-specific when $K_{t+1}$ is diagonal", has **not been mentioned** in this referenced publication. As such, we suggest that
> > - [2] focuses on scene-specific adaptations, and therefore, this is consistent with our argument on novelty: "Individual feedback is agent-specific, which differs from the scene-specific adaptation in [1,2]."
> > - The value of our work should not be penalized based on a viewpoint that has not been peer-reviewed or published.
> >
> > In addition, we would be very grateful if the reviewer could share his/her opinion on *how to make $K_{t+1}$ diagonal in practice, given that it is calculated as $K_{t+1} = S_{t+1|t}\Phi_{t+1}^\top P_{t+1}^{-1}$ in equation (3) of [2]*.
> >
> > > 2: In [2], one only needs to update the last layer of the neural network. There is no backpropagation happening onboard in [2].
> >
> > We highlight that
> > - As the reviewer stated, [2] updates the last layer of the neural network.
> > - As stated in our previous response, our proposed IAN works in a very different way from [2] and does not update parameters using data from the test scene.
> >
> > This is in fact a clear indication of the essential differences between our approach and [2], proving our novelty.
> >
> > It has come to our attention that the reviewer highlighted the sentence "there is no backpropagation happening onboard in [2]" in the comments. And we would like to clarify that we **did not use the term 'backpropagation'** in our previous response. What we actually wrote was '**update its** (the prediction model's) **parameters**', which is consistent with [2].
> >
> > ***
> >
> > As the discussion period is coming to an end, please do not hesitate to let us know if there are any further questions and we are glad to answer them.

---

> > ### Author Response · Authors · 2024-12-04
> > **Thank You for Your Efforts in Reviewing Our Submission**
> >
> > Dear Reviewer ykSs,
> >
> > As the discussion period has come to an end, we would like to express our gratitude for the time you devoted to reviewing and discussing our paper. While it appears that we had some differences of opinion regarding the novelty of our work, we sincerely hope you find our responses to your latest comments helpful.
> >
> > Best regards,
> >
> > Authors of Submission 7210

---

### Meta-Review · Area_Chair_CMSG · 2024-12-22

**Metareview:**

This paper investigates the human trajectory prediction via the perspective of individual feedback. Specifically, authors propose interactive adjustment network to model and leverage the feedback. Experiments look promising on several benchmarks. The major points to be summarized are as follows:

Positive:

+ easy to follow / good motivation
+ comprehensive experiments

Negative:

- Lack of some theoretical justification / missing technical details
- Elaboration on the key module, IAN
- Additional ablations / evaluation
- Unclear writing / typos

After rebuttal, authors did a good job to address most of the concerns. Most of the reviewers (4), out of 5 in total, are convincing that the manuscript is polished to great extent. All key issues (e.g. W1 by Reviewer dXFb on the assumption of ground truth, novelty compared to [1,2] by Reviewer ykSs, addiontal experiments Waymo, etc.) have been well addressed. In AC's view, the overall good merits over-weigh the downside of the manuscript. Please incorporate all the comments and elevate the paper for camera-ready version.

**Additional Comments On Reviewer Discussion:**

During rebuttal, there are sufficient improvement provided by the authors. In particular, there remains one reviewer that holds a negative rating on the manuscript. The core concern is the differences (thus novelty) between this method with [1,2], esp [2]. AC has read the orginal paper [2], there is no clear or evident overlap with this paper to [2].

---

### Decision · Program_Chairs · 2025-01-22

Accept (Poster)